# ZEROTH-ORDER METHODS FOR STOCHASTIC NONCONVEX NONSMOOTH COMPOSITE OPTIMIZATION

## ABSTRACT

This work aims to solve a stochastic nonconvex nonsmooth composite optimization problem. Previous works on composite optimization problem requires the major part to satisfy Lipschitz smoothness or some relaxed smoothness conditions, which excludes some machine learning examples such as regularized ReLU network and sparse support matrix machine. In this work, we focus on stochastic nonconvex composite optimization problem without any smoothness assumptions. In particular, we propose two new notions of approximate stationary points for such optimization problem and obtain finite-time convergence results of two zeroth-order algorithms to these two approximate stationary points respectively. Finally, we demonstrate that these algorithms are effective using numerical experiments.

## 1 INTRODUCTION

This work focuses on the following stochastic nonconvex nonsmooth composite optimization problem.

$$\min_{x \in \mathbb{R}^d} \phi(x) := F(x) + h(x), \text{ where } F(x) = \mathbb{E}_{\xi \sim \mathcal{P}}[f_\xi(x)], \tag{1}$$

where the individual function $f_\xi(x)$ is nonconvex and nonsmooth associated with a stochastic sample $\xi$ from the distribution $\mathcal{P}$, and $h$ is a convex regularizer. This problem covers many machine learning examples such as regularized ReLU network (Mazumdar & Rawat, 2019; Wang et al., 2021b) and sparse support matrix machine (Zheng et al., 2018; Gu et al., 2021; Li et al., 2022).

Existing approaches to the stochastic nonconvex composite optimization problem (1) require the major part $F$ to satisfy either Lipschitz smooth conditions (Nitanda, 2014; Li & Lin, 2015; Ghadimi et al., 2016; Ghadimi & Lan, 2016; Li et al., 2017; Pham et al., 2020), or some relaxed notions of smoothness such as relative smoothness (Bauschke et al., 2017; Lu et al., 2018; Latafat et al., 2022), smooth adaptivity (Wang & Han, 2023; Ding et al., 2025), anisotropic smoothness (Laude & Patrinos, 2025), weak convexity (Davis & Drusvyatskiy, 2019; Davis & Grimmer, 2019) and Holder continuous gradient (Guo et al., 2022), which cannot cover the applications with discontinous gradient, such as regularized ReLU network (Mazumdar & Rawat, 2019; Wang et al., 2021b) and sparse support matrix machine (Zheng et al., 2018; Gu et al., 2021; Li et al., 2022).

To solve such a stochastic nonconvex nonsmooth composite optimization problem, the first challenging step is to propose proper and feasible convergence criteria. The existing notions of proximal gradient mapping (Ghadimi et al., 2016; Reddi et al., 2016; Li & Li, 2018) and Frank-Wolfe gap (Jiang & Zhang, 2014; Guo et al., 2022) requiring $F$ to be differentiable everywhere are not suitable for nonsmooth composite optimization. Even after extending the gradient to the Clarke subdifferential, we will prove that convergence under the corresponding generalized stationary notions is intractable (see Theorem 1). Fortunately, Zhang et al. (2020) proposes the notion of $(\delta, \epsilon)$-Goldstein stationary point which has been achieved by various nonconvex nonsmooth optimization algorithms (Zhang et al., 2020; Lin et al., 2022; Chen et al., 2023; Cutkosky et al., 2023; Kornowski & Shamir, 2024), and the Goldstein stationary notion is extended to nonconvex nonsmooth constrained optimization (Liu et al., 2024). Inspired by these stationary notions, we propose $(\gamma, \delta, \epsilon)$-proximal Goldstein stationary point (PGSP) and $(\delta, \epsilon)$-conditional gradient Goldstein stationary point (CGGSP) as the approximate notions of stationarity for our nonconvex nonsmooth composite optimization problem (1), by using the Goldstein $\delta$-subdifferential (Goldstein, 1977) as a convex combination of the gradients in the neighborhood around the point of interest.

Table 1: Function evaluation complexity results of zeroth-order proximal gradient descent (0-PGD) and zeroth-order generalized conditional gradient algorithms (0-GCG).

| Main algorithm | Gradient estimation | Criterion | Complexity | Reference |
|---|---|---|---|---|
| 0-PGD (Algorithm 1) | Minibatch | $(\gamma, \delta, \epsilon)$-PGSP | $\mathcal{O}(d^{3/2}\delta^{-1}\epsilon^{-4})$ | Theorem 2 |
| 0-PGD (Algorithm 1) | Variance reduction | $(\gamma, \delta, \epsilon)$-PGSP | $\mathcal{O}(d^{3/2}\delta^{-1}\epsilon^{-3})$ | Theorem 3 |
| 0-GCG (Algorithm 2) | Minibatch | $(\delta, \epsilon)$-CGGSP | $\mathcal{O}(d^{3/2}\delta^{-1}\epsilon^{-4})$ | Theorem 4 |
| 0-GCG (Algorithm 2) | Variance reduction | $(\delta, \epsilon)$-CGGSP | $\mathcal{O}(d^{3/2}\delta^{-1}\epsilon^{-3})$ | Theorem 5 |

Using our proposed stationary notions above, we prove that the zeroth-order proximal gradient descent algorithm (0-PGD, see Algorithm 1) converges to our proposed $(\gamma, \delta, \epsilon)$-proximal Goldstein stationary point (PGSP), obtain the convergence rate and function evaluation complexity result using minibatch zeroth-order gradient estimation, and then improve these results using variance-reduced gradient estimation. Furthermore, we study a zeroth-order generalized conditional gradient algorithm (0-GCG, Algorithm 2) which avoids the possibly expensive proximal operator used by 0-PGD, and obtain similar convergence rate and complexity result of 0-GCG to achieve our proposed $(\delta, \epsilon)$-CGGSP. We summarize these convergence results of both algorithms in Table 1.

### 1.1 PAPER ORGANIZATION

Section 2 introduces the basic backgrounds including problem formulation, fundamentals for nonsmooth analysis and zeroth-order gradient estimation. Section 3 proposes our generalized stationary notions for composite optimization. Section 4 presents our zeroth-order proximal gradient descent (0-PGD) algorithm and its finite-time convergence results. Section 5 presents our zeroth-order generalized conditional gradient (0-GCG) algorithm and its finite-time convergence results. Section 6 shows the experimental results. Section 7 concludes this work.

## 2 PRELIMINARIES

In this section, we will introduce the problem formulation (Section 2.1), review fundamentals of nonsmooth analysis (Section 2.2), and introduce zeroth-order gradient estimation (Section 2.3).

### 2.1 PROBLEM FORMULATION

Throughout this work, we make the following two standard assumptions on the stochastic nonconvex nonsmooth composite optimization problem (1).

**Assumption 1.** *For any stochastic sample $\xi$, $f_\xi(x) : \mathbb{R}^d \to \mathbb{R}$ is an $L_\xi$-Lipschitz continuous for some $L_\xi > 0$ (i.e., $|f_\xi(y) - f_\xi(x)| \leq L_\xi \|y - x\|$ for any $x, y \in \mathbb{R}^d$) and $\mathbb{E}_\xi(L_\xi^2) \leq G^2$ for some $G > 0$.*

**Assumption 2.** *$h : \mathbb{R}^d \to \mathbb{R}$ is a proper closed convex function with at least one feasible point $x^{(h)} \in \mathbb{R}^d$ such that $h(x^{(h)}) < +\infty$.*

**Assumption 3.** *There exists $R > 0$ such that $h(x) > h(x^{(h)}) + G\|x - x^{(h)}\|$ for the feasible point $x^{(h)}$ defined in Assumption 2 and any $x \in \mathbb{R}^d$ satisfying $\|x - x^{(h)}\| > R$.*

Assumption 1 has also been used by (Davis & Drusvyatskiy, 2019; Davis & Grimmer, 2019; Liu et al., 2024). It implies that $F(x) = \mathbb{E}_\xi[f_\xi(x)]$ is $G$-Lipschitz continuous[1]. Such Lipschitz continuous but possibly nonsmooth functions have been widely used in optimization and machine learning, including any neural networks with ReLU activation (Krizhevsky et al., 2017; Mazumdar & Rawat, 2019; Ghosh et al., 2024; Shen et al., 2024), ramp loss (Gu et al., 2021; Wang & Shao, 2024), capped $\ell_1$ penalty (Xu et al., 2014; Zhang, 2008; Kumar et al., 2021), etc. Many commonly used convex regularizers $h$ satisfy Assumptions 2 and 3, including $\ell_p$ regularizer with $p > 1$ (McCulloch et al., 2024; Lu et al., 2024), $\ell_1$ regularizer $\lambda\|x\|_1$ with $\lambda > G$ to induce the sparsity of the parameter vector $x$ (Mazumdar & Rawat, 2019; Ali et al., 2024), super-coercive regularizer satisfying $\lim_{\|x\|\to+\infty}[h(x)/\|x\|] = +\infty$

---

[1]Assumption 1 implies that $F$ is $G$-Lipschitz continuous because for any $x, x' \in \mathbb{R}^d$,
$$|F(x') - F(x)| \leq \mathbb{E}_\xi|f_\xi(x') - f_\xi(x)| \leq \mathbb{E}_\xi[L_\xi\|x' - x\|] \leq \|x' - x\|\sqrt{\mathbb{E}_\xi[L_\xi^2]}.$$

(Bredies et al., 2005; Yu et al., 2017; Bredies et al., 2009), and the following constraint regularizer which enforces the constraint $x \in \Omega$ where $\Omega \subset \mathbb{R}^d$ is a convex and compact set (Jaggi, 2013; Rakotomamonjy et al., 2015; Nesterov, 2018; Liu et al., 2024; Assunção et al., 2025).

$$h_\Omega(x) \stackrel{\text{def}}{=} \begin{cases} 0; & x \in \Omega \\ +\infty; & x \notin \Omega \end{cases}, \tag{2}$$

**Proposition 1.** *Under Assumptions 1-3, the original objective function (1) has a non-empty solution set* $\arg\min_{x \in \mathbb{R}^d} \phi(x)$, *which is a subset of* $\mathcal{B}_d(x^{(h)}, R) \stackrel{\text{def}}{=} \{x \in \mathbb{R}^d : \|x - x^{(h)}\| \le R\}$.

**Remark:** Assumption 3 requires $h(x)$ to outgrow $F(x)$ as $\|x - x^{(h)}\| \to +\infty$, such that the objective $\phi(x) = F(x) + g(x)$ has minimizers and they are not too far from the feasible point $x^{(h)}$.

## 2.2 FUNDAMENTALS OF NONSMOOTH ANALYSIS

In this subsection, we will introduce some basic concepts for the unconstrained nonconvex nonsmooth optimization problem $\min_{x \in \mathbb{R}^d} F(x)$, a special case of the composite problem (1) with $h = 0$.

For a nondifferentiable function $F : \mathbb{R}^d \to \mathbb{R}$, we can define generalized directional derivatives and generalized gradients as follows.

**Definition 1.** *The **generalized directional derivative** of a function $F$ at point $x \in \mathbb{R}^d$ and direction $v \in \mathbb{R}^d$ is defined as* $DF(x; v) \stackrel{\text{def}}{=} \limsup_{y \to x, t \downarrow 0} \frac{F(x+tv) - F(x)}{t}$. *The Clark subdifferential of $F$ is defined as the set* $\partial F(x) \stackrel{\text{def}}{=} \{g \in \mathbb{R}^d : \langle g, v \rangle \le DF(x; v), \forall v \in \mathbb{R}^d\}$.

For the unconstrained nonconvex nonsmooth optimization problem $\min_{x \in \mathbb{R}^d} F(x)$, one may aim to find an $\epsilon$-Clarke stationary point defined as $x \in \mathbb{R}^d$ satisfying $\min\{\|g\| : g \in \partial F(x)\} \le \epsilon$. However, Zhang et al. (2020) proves that such an $\epsilon$-Clarke stationary point cannot be obtained in finite time for general Lipschitz continuous function $F$. Hence, they focus on more tractable and relaxed concepts of subdifferential and stationary solution, as defined below.

**Definition 2** (Goldstein (1977)). *The **Goldstein $\delta$-subdifferential** of a function $F$ at $x \in \mathbb{R}^d$ with radius $\delta \ge 0$ is defined as*

$$\partial_\delta F(x) \stackrel{\text{def}}{=} \text{conv}\big[ \cup_{y \in \mathcal{B}_d(x, \delta)} \partial F(y) \big],$$

*where $\text{conv}(A)$ denotes the set of every convex combination of the elements in $A$.*

**Definition 3** (Zhang et al. (2020)). *For any $\delta \ge 0$ and $\epsilon > 0$, a $(\delta, \epsilon)$-**Goldstein stationary point** of $F$ is defined as any $x \in \mathbb{R}^d$ satisfying*

$$\min\{\|g\| : g \in \partial_\delta F(x)\} \le \epsilon. \tag{3}$$

Note that $\partial_0 F(x) = \partial F(x)$ (Makela & Neittaanmaki, 1992). Hence, as $\delta = 0$, Goldstein $\delta$-subdifferential and $(\delta, \epsilon)$-Goldstein stationary point respectively reduce to Clark subdifferential and $\epsilon$-Clarke stationary point. Such a $(\delta, \epsilon)$-Goldstein stationary point can be achieved at finite time by various algorithms (Zhang et al., 2020; Tian et al., 2022; Davis et al., 2022; Cutkosky et al., 2023).

## 2.3 ZEROTH-ORDER GRADIENT ESTIMATION

Zeroth-order gradient estimation with random smoothing technique has been widely used when direct computation of gradient is costly or impossible. To estimate the gradient of a function $F$, we can approximate $F$ by its smoothing function $F_\delta(x) = \mathbb{E}_{u \sim Q}[F(x + \delta u)]$ with a small radius $\delta > 0$, with a certain distribution $Q$. We focus on the case where $Q$ is uniform distribution on the unit sphere $\mathcal{S}_d(1)) \stackrel{\text{def}}{=} \{u \in \mathbb{R}^d : \|u\| = 1\}$ (Duchi et al., 2015; Lin et al., 2020), since the corresponding smoothing function $F_\delta(x) \stackrel{\text{def}}{=} \mathbb{E}_{u \sim \text{Uniform}(\mathcal{S}_d(1))} F(x + \delta u)$ has the following amenable properties.

**Lemma 1** (Proposition 2.3 of (Lin et al., 2022)). *For any $G$-Lipschitz continuous function $F$, its smoothing function $F_\delta(x) \stackrel{\text{def}}{=} \mathbb{E}_{u \sim \text{Uniform}(\mathcal{S}_d(1))} F(x + \delta u)$ satisfies: (1) $\sup_{x \in \mathbb{R}^d} |F_\delta(x) - F(x)| \le \delta G$; (2) $F_\delta$ is $G$-Lipschitz continuous and differentiable everywhere with $cG\sqrt{d}/\delta$-Lipschitz continuous gradient for an absolute constant $c > 0$; (3) $\nabla F_\delta(x) \in \partial_\delta F(x)$ for any $x \in \mathbb{R}^d$.*

$\nabla F_\delta(x)$ admits the following unbiased two-point estimator, which is widely used in zeroth-order gradient estimation (Duchi et al., 2015; Lin et al., 2022; Ma & Huang, 2025).

$$\hat{g}_\delta(x; u, \xi) = \frac{d}{2\delta}[f_\xi(x + \delta u) - f_\xi(x - \delta u)]u, \tag{4}$$

where $\xi \sim \mathcal{P}$ and $u \sim \mathrm{Uniform}(\mathcal{S}_d(1))$.

# 3 GENERALIZED GOLDSTEIN STATIONARY POINTS FOR COMPOSITE OPTIMIZATION

In this section, we propose two new notions of stationary points for the stochastic nonconvex nonsmooth composite optimization problem (1), proximal Goldstein stationary point (PGSP) and conditional gradient Goldstein stationary point (CGGSP), the targets of zeroth-order algorithms in Sections 4 and 5. Then we show some properties of these new stationary points.

## 3.1 PROXIMAL GOLDSTEIN STATIONARY POINT (PGSP)

**Definition 4.** *For any stepsize $\gamma > 0$ and convex regularizer $h : \mathbb{R}^d \to \mathbb{R}$, we define the **proximal operator** of $\gamma h$ at point $x \in \mathbb{R}^d$ as follows (Parikh et al., 2014; Nitanda, 2014; Mardani et al., 2018; Yang & Yu, 2020).*

$$\mathrm{prox}_{\gamma h}(x) = \arg\min_{y \in \mathbb{R}^n}\left[h(y) + \frac{1}{2\gamma}\|y - x\|^2\right]. \tag{5}$$

The proximal operator (5) returns a unique solution since $h(y) + \frac{1}{2\gamma}\|y - x\|^2$ is a strongly convex function of $y$. Proximal operator is essential in the popular proximal gradient descent algorithms for composite optimization (Parikh et al., 2014; Nitanda, 2014; Mardani et al., 2018; Yang & Yu, 2020). We will use the proximal operator to propose a generalized notion of stationary point as follows.

**Definition 5.** *For any stepsize $\gamma > 0$ and convex regularizer $h : \mathbb{R}^d \to \mathbb{R}$, we define the **proximal gradient mapping** at point $x \in \mathbb{R}^d$ and gradient $g \in \mathbb{R}^d$ as follows (Ghadimi et al., 2016; Reddi et al., 2016; Li & Li, 2018).*

$$\mathcal{G}_{\gamma h}(x, g) = \frac{1}{\gamma}[x - \mathrm{prox}_{\gamma h}(x - \gamma g)]. \tag{6}$$

*Furthermore, for any $\epsilon \geq 0$, we define $x \in \mathbb{R}^d$ as a $(\gamma, \delta, \epsilon)$-**proximal Goldstein stationary point (PGSP)** if $\min_{g \in \partial_\delta F(x)} \|\mathcal{G}_{\gamma h}(x, g)\| \leq \epsilon$. Specifically, we call a $(\gamma, 0, \epsilon)$-PGSP as $(\gamma, \epsilon)$-**PGSP**, defined by $x \in \mathbb{R}^d$ such that $\min_{g \in \partial F(x)} \|\mathcal{G}_{\gamma h}(x, g)\| \leq \epsilon$.*

Our proposed notions of PGSP for nonsmooth composite optimization problem generalize existing stationary notions for the following special cases.

• For constrained optimization problem $\min_{x \in \Omega} F(x)$, a special case of the nonconvex nonsmooth composite optimization problem (1) with $h = h_\Omega$ defined by Eq. (2), $(\gamma, \delta, \epsilon)$-PGSP reduces to the $(\gamma, \delta, \epsilon)$-generalized Goldstein stationary point (Liu et al., 2024), where the proximal operator is reduced to the projection onto $\Omega$. Furthermore, when $h = 0$, $(\gamma, \delta, \epsilon)$-PGSP reduces to $(\delta, \epsilon)$-Goldstein stationary point (see Definition 3) (Zhang et al., 2020).

• When $F$ is differentiable, $(\gamma, \epsilon)$-PGSP has simplified definition that $\|\mathcal{G}_{\gamma h}[x, \nabla F(x)]\| \leq \epsilon^2$, which can be achieved by proximal gradient descent algorithms within finite-time (Ghadimi et al., 2016; Reddi et al., 2016; Li & Li, 2018). Furthermore, when $h = 0$, $(\gamma, \epsilon)$-PGSP reduces to $\epsilon$-stationary point defined as $x \in \mathbb{R}^d$ satisfying $\|\mathcal{G}_{\gamma h}[x, \nabla F(x)]\| \leq \epsilon$ which can be also achieved in finite time by many first-order algorithms. In contrast, $(\gamma, \epsilon)$-PGSP is intractable for our setting with Lipschitz continuous and nondifferentiable $F$, as will be shown later in Theorem 1. Therefore, we aim at $(\gamma, \delta, \epsilon)$-PGSP, a relaxed notion of stationarity, and will propose a zeroth-order proximal gradient descent algorithm (Algorithm 1) that achieves this point in finite time.

Our proposed notions of PGSP satisfy the following properties.

---

[2] For differentiable function $F$, we have $\partial F(x) = \{\nabla F(x)\}$ (Makela & Neittaanmaki, 1992).

**Proposition 2.** *Suppose the function $F : \mathbb{R}^d \to \mathbb{R}$ is differentiable and $h : \mathbb{R}^d \to \mathbb{R}$ is a convex function. Then, $(\gamma, \delta, \epsilon)$-PGSP has the following properties.*

*1. A $(\gamma, \epsilon)$-PGSP is also a $(\gamma, \delta, \epsilon)$-PGSP.*

*2. If $\nabla F$ is $L$-Lipschitz continuous, then a $(\gamma, \epsilon/(2L), \epsilon/2)$-PGSP is also a $(\gamma, \epsilon)$-PGSP, i.e., $\|\mathcal{G}_{\gamma h}[x, \nabla F(x)]\| \leq \epsilon$.*

*3. If $x \in \mathbb{R}^d$ satisfies $\|\mathcal{G}_{\gamma h}[x, \nabla F_\delta(x)]\| \leq \epsilon$, then $x$ is a $(\gamma, \delta, \epsilon)$-PGSP.*

**Remark:** In Proposition 2, items 1 and 2 imply that for $L$-Lipschitz smooth functions $F$, our notions of $(\gamma, \delta, \epsilon)$-PGSP and $(\gamma, \epsilon')$-PGSP are equivalent (for possibly different $\epsilon, \epsilon' \geq 0$). Item 3 implies that we can obtain a $(\gamma, \delta, \epsilon)$-PGSP by solving $\min_{x \in \mathbb{R}^d} F_\delta(x)$, which is important for designing the zeroth-order proximal gradient descent algorithm (Algorithm 1).

### 3.2 Conditional Gradient Goldstein Stationary Point (CGGSP)

Note that the PGSP defined in the previous subsection relies on the stepsize $\gamma$. In this subsection, we will define conditional gradient Goldstein stationary point (CGGSP), a notion of stationary point that does not rely on the stepsize $\gamma$ and can be computationally cheaper.

**Definition 6.** *For any convex regularizer $h : \mathbb{R}^d \to \mathbb{R}$, we define the **linear minimization oracle (LMO)** of $h$ at the gradient $g \in \mathbb{R}^d$ as follows.*

$$\mathcal{L}_h(g) \stackrel{\text{def}}{=} \arg\min_{y \in \mathbb{R}^d} \left[ h(y) + \langle y, g \rangle \right]. \tag{7}$$

The LMO defined above always exist and is bounded as shown below.

**Proposition 3.** *Under Assumptions 2-3, for any $\|g\| \leq G$, the LMO (7) yields a non-empty set $\mathcal{L}_h(g) \subset \mathcal{B}_d(x^{(h)}, R) \stackrel{\text{def}}{=} \{x \in \mathbb{R}^d : \|x - x^{(h)}\| \leq R\}$.*

Linear minimization oracle (LMO) has been adopted to develop generalized conditional gradient methods for composite optimization (Jiang & Zhang, 2014; Ghadimi, 2019). In the constrained optimization $\min_{x \in \Omega} F(x)$ as a special case, LMO reduces to $\arg\min_{y \in \Omega} \langle y, g \rangle$ used by the Frank-Wolfe algorithm (Frank et al., 1956; Lan & Zhou, 2016). Compared with the proximal operator (5), LMO can be computationally cheaper Juditsky & Nemirovski (2016). We will also use LMO to propose a computationally cheaper notion of stationary point as follows.

**Definition 7.** *For any convex regularizer $h : \mathbb{R}^d \to \mathbb{R}$, we define the $\delta$-**regularized Frank-Wolfe gap** of $h$ at point $x \in \mathbb{R}^d$ and gradient $g \in \mathbb{R}^d$ as follows.*

$$\mathcal{W}_h(x, g) \stackrel{\text{def}}{=} \max_{y \in \mathbb{R}^d} \left[ h(x) - h(y) + \langle y - x, -g \rangle \right] \stackrel{\text{Eq.(7)}}{=} h(x) - h(y') + \langle y' - x, -g \rangle, \tag{8}$$

*for any $y' \in \mathcal{L}_h(g)$. Furthermore, for any $\epsilon \geq 0$, we define $x \in \mathbb{R}^d$ as a $(\delta, \epsilon)$-**conditional gradient Goldstein stationary point (CGGSP)** if $\min_{g \in \partial_\delta F(x)} \mathcal{W}_h(x, g) \leq \epsilon$. Specifically, a $(0, \epsilon)$-CGGSP is also called an $\epsilon$-CGGSP, defined by $x \in \mathbb{R}^d$ such that $\min_{g \in \partial F(x)} \mathcal{W}_h(x, g) \leq \epsilon$.*

Our proposed notions of CGGSP for nonsmooth composite optimization problem generalizes existing stationary notions for the following special cases.

• For constrained optimization problem $\min_{x \in \Omega} F(x)$, a special case of the nonconvex nonsmooth composite optimization problem (1) with $h = h_\Omega$ defined by Eq. (2), $(\delta, \epsilon)$-CGGSP reduces to the $(\delta, \epsilon)$-Clarke Frank-Wolfe stationary point (Liu et al., 2024).

• When $F$ is differentiable, $\epsilon$-CGGSP has simplified definition that $\|\mathcal{W}_h[x, \nabla F(x)]\| \leq \epsilon^3$, which can be achieved by conditional gradient descent algorithms within finite-time (Jiang & Zhang, 2014; Guo et al., 2022). In contrast, $(\gamma, \epsilon)$-CGGSP is intractable for our setting with Lipschitz continuous and nondifferentiable $F$, as will be shown later in Theorem 1. Therefore, we aim at $(\delta, \epsilon)$-CGGSP, a relaxed notion of stationarity, and will propose a zeroth-order generalized conditional gradient algorithm (Algorithm 2) that achieves this point in finite time.

Our proposed notions of CGGSP satisfy the following properties.

---

[3] For differentiable function $F$, we have $\partial F(x) = \{\nabla F(x)\}$ (Makela & Neittaanmaki, 1992).

**Proposition 4.** *Suppose the function $F : \mathbb{R}^d \to \mathbb{R}$ is differentiable and $h : \mathbb{R}^d \to \mathbb{R}$ satisfies Assumption 2. Then $(\delta, \epsilon)$-CGGSP has the following properties.*
*1. An $\epsilon$-CGGSP is also a $(\delta, \epsilon)$-CGGSP.*
*2. Suppose $\nabla F$ is L-Lipschitz continuous. Then a $(\epsilon/(2RL), \epsilon/2)$-CGGSP is also an $\epsilon$-CGGSP, i.e.,*
*$\mathcal{W}_h[x, \nabla F(x)] \le \epsilon$.*
*3. If $x \in \mathbb{R}^d$ satisfies $\mathcal{W}_h[x, \nabla F_\delta(x)] \le \epsilon$, then $x$ is a $(\delta, \epsilon)$-CGGSP.*

**Remark:** Items 1-3 of Proposition 4 for CGGSP are analogous to items 1-3 of Proposition 2 for PGSP. Specifically, items 1 and 2 imply that for $L$-Lipschitz smooth functions $F$, our notions of $(\delta, \epsilon)$-CGGSP and $\epsilon'$-CGGSP are equivalent (for possibly different $\epsilon, \epsilon' \ge 0$). Item 3 implies that we can obtain a $(\delta, \epsilon)$-CGGSP by solving $\min_{x \in \mathbb{R}^d} F_\delta(x)$, which is important for later designing the zeroth-order generalized conditional gradient algorithm (Algorithm 2).

Finally, Theorem 1 below shows that for composite problem (1) with general nonsmooth Lipschitz continuous function $F$, $(\gamma, \epsilon)$-PGSP and $\epsilon$-CGGSP are intractable. In contrast, $(\gamma, \delta, \epsilon)$-PGSP and $(\delta, \epsilon)$-CGGSP can be achieved in finite time, by two zeroth-order algorithms presented in the two consequent sections respectively.

**Theorem 1.** *Consider any $T \in \mathbb{N}$, $d \ge 2$ and any randomized algorithm $\mathcal{A}$ with access to a local oracle of the objective function (1) [4] Then there exist functions $F$ and $h$ satisfying Assumptions 1-2 such that $\phi(0) - \inf_{x \in \mathbb{R}^d} \phi(x) \le 2$ but with probability at least $1 - 2T \exp(-d/36)$, none of $\{x_t\}_{t=1}^T$ generalized by $\mathcal{A}$ belongs to the set of $(\gamma, \epsilon)$-PGSP or $\epsilon$-CGGSP for $\epsilon \in \left(0, \frac{1}{4\sqrt{2}}\right)$ and $\gamma \in (0, 0.1]$.*

# 4 ZEROTH-ORDER PROXIMAL GRADIENT DESCENT (0-PGD) ALGORITHM

In this section we study a zeroth-order proximal gradient descent (0-PGD) algorithm, as shown in Algorithm 1. The main algorithm framework is proximal gradient descent update (2) on the composite optimization problem $\min_{x \in \mathbb{R}^d}[F_\delta(x) + h(x)]$ that approximates the original problem (1), where the zeroth-order stochastic gradient estimator $g_t \approx \nabla F_\delta(x_t)$ is obtained using either minibatch estimation (option G1) or variance-reduced estimation (option G2).

We first present the convergence results of Algorithm 1 with minibatch estimation as follows.

**Theorem 2** (Convergence of 0-PGD Algorithm with Minibatch Gradient Estimation)**.** *Implement Algorithm 1 with Option G1, stepsize $\gamma = \frac{\delta}{cG\sqrt{d}}$ and constant batchsize $B_t \equiv B$. Then under Assumptions 1-2, the output $x_{\widetilde{T}}$ has the following convergence rate.*

$$\mathbb{E}[\|\mathcal{G}_{\gamma h}(x_{\widetilde{T}}, \nabla F_\delta(x_{\widetilde{T}})\|] \le \frac{\sqrt{2cG}d^{1/4}}{\sqrt{T\delta}}\sqrt{\mathbb{E}[\phi(x_0)] - \phi_{\min} + 2\delta G} + \frac{16G\sqrt{d}}{\sqrt{B}} \tag{12}$$

*where $\phi_{\min} \overset{\text{def}}{=} \min_{x \in \mathbb{R}^d}[F(x) + h(x)]$. Furthermore, we can obtain a $(\gamma, \delta, \epsilon)$-PGSP by using hyperparameters $T = \mathcal{O}(Gd^{1/2}\delta^{-1}\epsilon^{-2})$, $B = \mathcal{O}(G^2 d\epsilon^{-2})$ (see their full expressions in Eqs. (35) and (36) in Appendix I), which requires at most $2TB = \mathcal{O}(G^3 d^{3/2}\delta^{-1}\epsilon^{-4})$ function evaluations and $T = \mathcal{O}(Gd^{1/2}\delta^{-1}\epsilon^{-2})$ proximal updates (2).*

**Comparison with Constrained Optimization:** The stochastic constrained optimization problem $\min_{x \in \Omega}\{F(x) \overset{\text{def}}{=} \mathbb{E}_\xi[F_\xi(x)]\}$ on a convex and compact set $\Omega$ is a special case of the composition optimization problem (1) by using $h = h_\Omega$ defined in Eq. (2). Liu et al. (2024) studies this constrained optimization with also nonconvex, nonsmooth and $G$-Lipschitz continuous $F$, proposes a stochastic projected gradient descent algorithm as a special case of our Algorithm 1, and obtains the function evaluation complexity result $\mathcal{O}(G^4 Rd^{3/2}\delta^{-1}\epsilon^{-4})$ to achieve a $(\gamma, \delta, \epsilon)$-generalized Goldstein stationary point as a special case of our $(\gamma, \delta, \epsilon)$-PGSP (see Corollary 5.2 of (Liu et al., 2024)). This complexity result requires $\Omega$ to be bounded with radius $R$ and is higher than our $\mathcal{O}(G^3 d^{3/2}\delta^{-1}\epsilon^{-4})$ that does not require $R$. Our improvement is obtained by replacing their bound $F_\delta(x_0) - F_\delta(x_T) \le G\|x_0 - x_T\| \le 2GR$ with the tighter bound $F_\delta(x_0) + h(x_0) - F_\delta(x_T) - h(x_T) \le \phi(x_0) - \phi_{\min} + 2\delta G$.

---

[4]A local oracle means a quantity $\mathcal{O}_{F,h}(x)$ (e.g. $F(x), \nabla F(x) + \partial h(x)$) that reveals local information about the function values of $F$ and $h$ around a certain point $x \in \mathbb{R}^d$.

---

**Algorithm 1** Zeroth-order proximal gradient descent (0-PGD) algorithm

---

1: **Inputs:** Number of iterations $T$, stepsize $\gamma > 0$, batchsizes $B_t$, radius $\delta > 0$, period $q$ for variance reduction.

2: **Initialize:** $x_0 \in \mathbb{R}^d$.

3: **for** iterations $t = 0, 1, \ldots, T - 1$ **do**

4:     Obtain i.i.d. samples $\{u_{i,t}\}_{i=1}^{B_t} \sim \text{Uniform}(\mathcal{S}_d(1))$ and $\{\xi_{i,t}\}_{i=1}^{B_t} \sim \mathcal{P}$.

5:     Obtain stochastic gradient estimation $g_t \approx \nabla F_\delta(x_t)$ by either option below.

6:

7:     **Option G1: Minibatch Estimation.**

$$g_t = \frac{1}{B_t} \sum_{i=1}^{B_t} \hat{g}_\delta(x_t, u_{i,t}, \xi_{i,t}), \tag{9}$$

    where $\hat{g}_\delta$ is defined by Eq. (4).

8:

9:     **Option G2: Variance-reduced Estimation.**

10:     **if** $t \bmod q = 0$ **then**

11:       Obtain $g_t$ by Eq. (9).

12:     **else**

13:       Obtain $g_t$ by the following variance-reduced estimation.

$$g_t = g_{t-1} + \frac{1}{B_t} \sum_{i=1}^{B_t} \left[ \hat{g}_\delta(x_t, u_{i,t}, \xi_{i,t}) - \hat{g}_\delta(x_{t-1}, u_{i,t}, \xi_{i,t}) \right], \tag{10}$$

      where $\hat{g}_\delta$ is defined by Eq. (4).

14:     **end if**

15:

16:     Update $x_t$ by proximal gradient descent as follows.

$$x_{t+1} = \text{prox}_{\gamma h}(x_t - \gamma g_t) \overset{\text{def}}{=} \arg\min_{y \in \mathbb{R}^n} \left[ h(y) + \frac{1}{2\gamma} \|y - x_t + \gamma g_t\|^2 \right], \tag{11}$$

    where the proximal operator $\text{prox}_{\gamma h}$ is defined by Eq. (5).

17: **end for**

18: **Output:** $x_{\widetilde{T}}$ where $\widetilde{T}$ is uniformly obtained from $\{0, 1, \ldots, T - 1\}$ at random.

---

Then using variance reduced gradient estimation, we obtain the following improved convergence rate and complexity results of Algorithms 1 as follows.

**Theorem 3** (Convergence of 0-PGD Algorithm with Variance Reduction). *Implement Algorithm 1 with Option G2, stepsize $\gamma = \frac{\delta}{2G(d+c\sqrt{d})}$, batchsize $B_t = B_0$ for any $t \bmod q = 0$ and $B_t = B_1 = q$ for other $t$. Then under Assumptions 1-2, the output $x_{\widetilde{T}}$ has the following convergence rate.*

$$\mathbb{E}[\|\mathcal{G}_{\gamma h}(x_{\widetilde{T}}, \nabla F_\delta(x_{\widetilde{T}})\|] \leq \frac{\sqrt{2cG}d^{1/4}}{\sqrt{T}\delta} \sqrt{\mathbb{E}[\phi(x_0)] - \phi_{\min} + 2\delta G} + \frac{16G\sqrt{d}}{\sqrt{B}} \tag{13}$$

*Furthermore, we can obtain a $(\gamma, \delta, \epsilon)$-PGSP by using hyperparameters $B_0 = 1764dG^2\epsilon^{-2}$, $B_1 = q = 42\sqrt{d}G\epsilon^{-1}$, $T = \mathcal{O}(Gd\delta^{-1}\epsilon^{-2})$ (see their full expressions in Eq. (39) in Appendix J), which requires at most $2B_0\lfloor T/q \rfloor + 4B_1(T - \lfloor T/q \rfloor) = \mathcal{O}(G^2d^{3/2}\delta^{-1}\epsilon^{-3})$ function evaluations and $T = \mathcal{O}(Gd\delta^{-1}\epsilon^{-2})$ proximal updates (2).*

**Comparison with Existing Results:** For stochastic nonconvex nonsmooth constrained optimization $\min_{x \in \Omega}\{F(x) \overset{\text{def}}{=} \mathbb{E}_\xi[F_\xi(x)]\}$ as a special case, Liu et al. (2024) obtains function evaluation complexity $\mathcal{O}(G^3Rd^{3/2}\delta^{-1}\epsilon^{-3})$, higher than our $\mathcal{O}(G^2d^{3/2}\delta^{-1}\epsilon^{-3})$ (see their Corollary 5.4). For unconstrained optimization $\min_{x \in \mathbb{R}^d} F(x)$, a smaller special case, Chen et al. (2023) uses variance reduction to achieve a $(\delta, \epsilon)$-Goldstein stationary point using variance reduction with also $\mathcal{O}(G^2d^{3/2}\delta^{-1}\epsilon^{-3})$ function evaluations that matches our complexity result (see their Theorem 1).

# 5 Zeroth-Order Generalized Conditional Gradient (0-GCG) Algorithm

In this section, we consider the case where the proximal operator (5) is costly. For example, when $h(x)$ is a nuclear norm of regularizer, the proximal operator requires full singular value decomposition (Wang et al., 2021a). The popular generalized conditional gradient method (Bredies et al., 2005; Jiang & Zhang, 2014; Rakotomamonjy et al., 2015; Bach, 2015; Nesterov, 2018; Ghadimi, 2019; Guo et al., 2022; Ito et al., 2023) uses a cheaper linear minimization oracle (LMO, defined by Eq. (7)) to replace the proximal operator. We propose a zeroth-order generalized conditional gradient method, using also two options of the zeroth-order gradient estimations, minibatch estimation (option G1) and variance-reduced estimation (option G2), as shown in Algorithm 2.

---

**Algorithm 2** Zeroth-order generalized conditional gradient algorithm (0-GCG)

---

1: **Inputs:** Number of iterations $T$, stepsize $\gamma > 0$, batchsizes $B_t$, radius $\delta > 0$, period $q$ for variance reduction.
2: **Initialize:** $x_0 \in \mathbb{R}^d$.
3: **for** iterations $t = 0, 1, \ldots, T - 1$ **do**
4:     Obtain i.i.d. samples $\{u_{i,t}\}_{i=1}^{B_t} \sim \mathrm{Uniform}(\mathcal{S}_d(1))$ and $\{\xi_{i,t}\}_{i=1}^{B_t} \sim \mathcal{P}$.
5:     Obtain stochastic gradient estimation $g_t \approx \nabla F_\delta(x_t)$ by option G1 or G2 in Algorithm 1.
6:     Update $x_t$ using LMO as follows.

$$y_t \in \mathcal{L}_h(g_t) \overset{\text{def}}{=} \arg\min_{y \in \mathbb{R}^d}[h(y) + \langle g_t, y\rangle], \quad (14)$$
$$x_{t+1} = x_t + \gamma(y_t - x_t). \quad (15)$$

7: **end for**
8: **Output:** $x_{\widetilde{T}}$ where $\widetilde{T}$ is uniformly obtained from $\{0, 1, \ldots, T - 1\}$ at random.

---

We present the convergence rate and complexity results of Algorithm 2 in the following two theorems, for the two gradient estimation options respectively.

**Theorem 4** (Convergence of 0-GCG Algorithm with Minibatch Gradient Estimation). *Implement Algorithm 2 with Option G1, stepsize $\gamma = \frac{1}{R}\sqrt{\frac{2\delta}{TcG\sqrt{d}}\mathbb{E}[\phi(x_0) - \phi_{\min} + 2\delta G]}$, constant batchsize $B_t \equiv B$ and initial point $x_0$ satisfying $\|x_0 - x^{(h)}\| \leq R$. Then under Assumptions 1-3, the output $x_{\widetilde{T}}$ has the following convergence rate.*

$$\mathbb{E}\big[\mathcal{W}_h[x_{\widetilde{T}}, \nabla F_\delta(x_{\widetilde{T}})]\big] \leq R\sqrt{\frac{8cG\sqrt{d}}{T\delta}\mathbb{E}[\phi(x_0) - \phi_{\min} + 2\delta G]} + \frac{21RG\sqrt{d}}{\sqrt{B}}. \quad (16)$$

*Furthermore, we can obtain a $(\delta, \epsilon)$-CGGSP by using hyperparameters $T = \mathcal{O}(GR^2d^{1/2}\delta^{-1}\epsilon^{-2})$ (see its full expression in Eq. (43) in Appendix K), $B = 1764G^2dR^2\epsilon^{-2}$, which requires at most $2TB = \mathcal{O}(G^3R^4d^{3/2}\delta^{-1}\epsilon^{-4})$ function evaluations and $T = \mathcal{O}(GR^2d^{1/2}\delta^{-1}\epsilon^{-2})$ LMO updates (14).*

**Theorem 5** (Convergence of 0-GCG Algorithm with Variance Reduction). *Implement Algorithm 2 with Option G2, stepsize $\gamma = \frac{1}{R}\sqrt{\frac{\delta\mathbb{E}[\phi(x_0) - \phi_{\min} + 2\delta G]}{TG(4d + 2c\sqrt{d})}}$, batchsize $B_t = B_0$ for any $t \bmod q = 0$ and $B_t = B_1 = q$ for other $t$. The initial point $x_0$ satisfies $\|x_0 - x^{(h)}\| \leq R$. Then under Assumptions 1-3, the output $x_{\widetilde{T}}$ has the following convergence rate.*

$$\mathbb{E}[\|\mathcal{W}_h(x_{\widetilde{T}}, \nabla F_\delta(x_{\widetilde{T}}))\|] \leq 2R\sqrt{\frac{G(4d + 2c\sqrt{d})}{T\delta}\mathbb{E}[\phi(x_0) - \phi_{\min} + 2\delta G]} + \frac{13RG\sqrt{d}}{\sqrt{B_0}} \quad (17)$$

*Furthermore, we can obtain a $(\delta, \epsilon)$-CGGSP by using hyperparameters $B_0 = 676dR^2G^2\epsilon^{-2}$, $B_1 = q = 26RG\epsilon^{-1}\sqrt{d}$, $T = \mathcal{O}(GR^2d\delta^{-1}\epsilon^{-2})$ (see its full expression in Eq. (45) in Appendix L), which requires at most $2B_0\lfloor T/q\rfloor + 4B_1(T - \lfloor T/q\rfloor) = \mathcal{O}(G^2R^3d^{3/2}\delta^{-1}\epsilon^{-3})$ function evaluations and $T = \mathcal{O}(GR^2d\delta^{-1}\epsilon^{-2})$ LMO updates (14).*

**Comparison with Constrained Optimization:** With minibatch gradient estimation, our function evaluation complexity $\mathcal{O}(G^3R^4d^{3/2}\delta^{-1}\epsilon^{-4})$ in Theorem 4 is more efficient than the complexity $\mathcal{O}(G^4R^5d^{3/2}\delta^{-1}\epsilon^{-4})$ to achieve $(\delta, \epsilon)$-Goldstein Frank–Wolfe stationary point of the stochastic nonconvex nonsmooth constrained optimization $\min_{x\in\Omega}\{F(x) \overset{\text{def}}{=} \mathbb{E}_\xi[F_\xi(x)]\}$, a special case of our $(\delta, \epsilon)$-CGGSP of the composite optimization problem (1) (Corollary 5.7 of (Liu et al., 2024)). Using variance reduction, our complexity improves to $\mathcal{O}(G^2R^3d^{3/2}\delta^{-1}\epsilon^{-3})$, which is also lower than $\mathcal{O}(G^3R^4d^{3/2}\delta^{-1}\epsilon^{-3})$ for $\min_{x\in\Omega}\{F(x) \overset{\text{def}}{=} \mathbb{E}_\xi[F_\xi(x)]\}$ (Corollary 5.9 of (Liu et al., 2024)).

## 6 EXPERIMENTS

We apply our zeroth-order algorithms to train a two-layer ReLu network $r_\xi(x) = W_2\sigma(W_1\xi+b_1)+b_2$. Here, $\xi \in \mathbb{R}^{d_\xi}$ is an input sample. The network parameters include the weight matrices ($W_1 \in \mathbb{R}^{d_1 \times d_\xi}$ and $W_2 \in \mathbb{R}^{d_2 \times d_1}$) and bias vectors ($b_1 \in \mathbb{R}^{d_1}$ and $b_2 \in \mathbb{R}^{d_2}$). $x \in \mathbb{R}^d$ denotes the total parameter which is concatenated by $b_1$, $b_2$ and flattened $W_1$, $W_2$, so the total dimensionality is $d = d_1d_\xi + d_1d_2 + d_1 + d_2$. $\sigma : \mathbb{R}^{d_1} \to \mathbb{R}^{d_1}$ is the widely used ReLu activation function which maps each entry $u$ to $\max(u, 0)$.

We select $d_\xi = 5$, $d_1 = 4$ and $d_2 = 2$ which imply $d = 34$, and generate the underlying sparse parameters $x^* \in \mathbb{R}^d$ by randomly selecting half of the entries to be 0 and generating the other half from standard Gaussian. Then we construct the binary classification dataset $\{(\xi_i, y_i)\}_{i=1}^N$ with sample size $N = 1000$, where the inputs $\xi_i \in \mathbb{R}^5$ are i.i.d. standard Gaussian, and the label $y_i = 0$ if the first entry of $f_{\xi_i}(x^*) \in \mathbb{R}^2$ is larger, otherwise $y_i = 1$. Then we train the regularized ReLu network via the following composite optimization problem.

$$\min_{x\in\mathbb{R}^d} \phi(x) = \frac{1}{N}\sum_{i=1}^N \ell[r_{\xi_i}(x), y_i] + \lambda_1\|x\|_1 + \frac{\lambda_2}{2}\|x\|_2^2, \tag{18}$$

This can be seen as an instance of the problem (1), where the main part $F(x) = \frac{1}{N}\sum_{i=1}^N \ell[r_{\xi_i}(x), y_i]$ denotes the average cross-entropy loss between the prediction $r_{\xi_i}(x)$ and the true label $y_i$, and is nonsmooth due to the ReLu activation $\sigma$. In the convex regularizer $h(x) = \lambda_1\|x\|_1 + \frac{\lambda_2}{2}\|x\|_2^2$, we select $\lambda_1 = \lambda_2 = 0.01$, $\|x\|_1$ induces sparse parameters and $\|x\|_2$ controls the parameter magnitude.

We implement our Algorithms 1 and 2, and for each algorithm we test both gradient estimation options, G1 (minibatch) and G2 (variance reduction), all with radius $\delta = 0.001$. For both algorithms with option G1 we select batchsize 500 and run 100 iterations. For both algorithms with option G2 we run 523 iterations, start each epoch of 10 iterations with batchsize 500, and use batchsize 50 for the rest iterations. We use fine-tuned stepsizes 0.005 for 0-PGD with G1, 0.001 for 0-PGD with G2, $5 \times 10^{-5}$ for 0-GCG with G1, and $10^{-5}$ for 0-GCG with G2. The experiment is conducted on Python 3.9 using Apple M1 Pro with 8 cores and 16 GB memory, which costs about half a minute.

At each iteration $t$, we evaluate the training objective function $\phi(x_t)$ (Eq. (18)) as well as the classification accuracies on both the 1000 training samples and the 1000 heldout test samples generated in the same way as that of the training samples. In Figure 1, we plot these metrics VS the function evaluation complexity (the total number of function evaluations up to each iteration), which shows that all the algorithms converge well with over 90% accuracy on both training and test samples. In particular, compared with minibatch gradient estimation (option G1), after improving gradient estimation with variance reduction (option G2), both algorithms 0-PGD and 0-GCG converge faster.

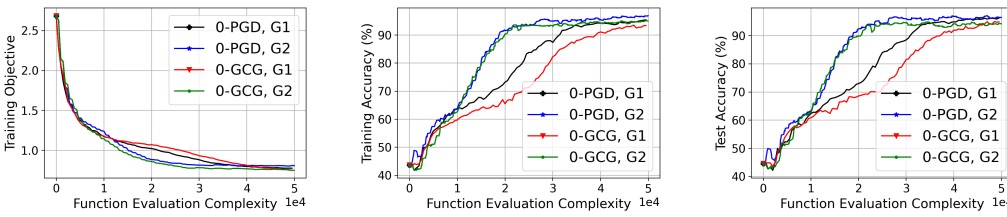

Figure 1: Experimental results on regularized ReLu network.

## 7 CONCLUSION

In this work, we have proposed two new notions of stationary points for stochastic nonconvex nonsmooth composite optimization, the $(\gamma, \delta, \epsilon)$-proximal Goldstein stationary point (PGSP) and the $(\delta, \epsilon)$-conditional gradient Goldstein stationary point (CGGSP). We have also proved that the zeroth-order proximal gradient descent algorithm (0-PGD) and the zeroth-order generalized conditional gradient algorithm (0-GCG) converge to a $(\gamma, \delta, \epsilon)$-PGSP and a $(\delta, \epsilon)$-CGGSP respectively, and obtained the convergence rates and complexity results. The experimental results on regularized ReLu network show that these algorithms converge well.

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

# Appendix

## Table of Contents

## A    RELATED WORKS

**Finite-time Convergence Results on Nonconvex Nonsmooth Optimization:** Only recently are finite-time convergence results obtained on nonconvex nonsmooth optimization, a special case of our nonconvex nonsmooth composite optimization (1). Davis & Drusvyatskiy (2019); Davis & Grimmer (2019) obtain finite-time convergence for stochastic optimization of a $\rho$-weakly convex function. Zhang et al. (2020) obtains the first dimension-free convergence result to achieve a $(\delta, \epsilon)$-Goldstein stationary point, which involves impractical subgradient computation. Such subgradient computation is removed by (Davis et al., 2022; Tian et al., 2022) using perturbations. Jordan et al. (2023); Tian & So (2024) prove that deterministic algorithms cannot obtain dimension-free convergence for nonconvex nonsmooth optimization. Cutkosky et al. (2023) obtains the optimal complexity result using online learning.

Nesterov & Spokoiny (2017) obtains the first finite-time convergence result of zeroth-order methods for stochastic nonconvex nonsmooth optimization. Lin et al. (2022) designs zeroth-order algorithms with provable finite-time convergence to $(\delta, \epsilon)$-Goldstein stationary point. Their oracle complexity is improved by (Chen et al., 2023) using variance reduction, and further improved to the optimal complexity $\mathcal{O}(d\delta^{-1}\epsilon^{-3})$ by (Kornowski & Shamir, 2024) using the online learning technique in (Cutkosky et al., 2023).

**Proximal Gradient Methods:** Various proximal gradient methods are very popular for various composite optimization problem (1). For example, Fukushima & Mine (1981) derives asymptotic convergence of proximal gradient method for smooth composite optimization problem [5] under

---

[5]Here, smooth composite optimization means the major part $F$ of the composite optimization problem (1) is Lipschitz smooth.

both convex and nonconvex settings. Attouch et al. (2013) analyzes the convergence of multiple variants of inexact proximal algorithms [6] on smooth nonconvex composite optimization satisfying Kurdyka–Łojasiewicz geometry. Ghadimi et al. (2016) analyzes the convergence of proximal gradient method for non-stochastic smooth composite optimization, and that of the two algorithm variants with minibatch and zeroth-order gradient estimations for stochastic smooth nonconvex and convex composite optimization problems. Beck & Teboulle (2009) applies Nesterov's acceleration to proximal gradient method[7] and uses backtracking technique to estimate Lipschitz constant, for smooth convex composite optimization. Nitanda (2014) combines Nesterov's acceleration and SVRG variance reduction in proximal gradient methods and thus improved convergence rate for stochastic smooth convex composite optimization. (Li & Lin, 2015; Ghadimi & Lan, 2016; Li et al., 2017) study accelerated proximal gradient (APG) algorithms for nonconvex smooth composite optimization. Stochastic proximal gradient methods have been accelerated by variance reduction techniques, such as SVRG (Reddi et al., 2016; Li et al., 2017), SAGA (Reddi et al., 2016), SARAH (Pham et al., 2020), and adaptive APG algorithm with Spider variance reduction. Proximal gradient methods have also been extended from Euclidean distance to Bregman distance. For example, Bregman distance based proximal gradient method has provable convergence results for Bregman distance based relatively smooth composite optimization under both convex (Bauschke et al., 2017) and nonconvex settings (Latafat et al., 2022; Wang & Han, 2023). Ding et al. (2025) obtains the optimal sample complexity results of both Bregman proximal gradient method and its momentum variant for smooth adaptable composite optimization. Laude & Patrinos (2025) analyzes an anisotropic proximal gradient method for anisotropic smooth composite optimization.

**Conditional Gradient Methods:** Frank et al. (1956) proposes conditional gradient method (also known as Frank-Wolfe algorithm) for quadratic programming. Lan & Zhou (2016) extends conditional gradient method to convex optimization by skipping gradient evaluations, and achieved optimal computation complexity results. Bredies et al. (2005) proposes a generalized conditional gradient method which extends to composite optimization, the focus of this work, and obtains asymptotic convergence result for nonconvex setting. Since then, generalized conditional gradient methods have been applied to various composite optimization problems. For example, Jiang & Zhang (2014) studies nonconvex and nonsmooth composite optimization with block-structure. Bach (2015); Nesterov (2018) focus on general convex composite optimization problems. Harchaoui et al. (2015) studies norm-regularized convex optimization. Rakotomamonjy et al. (2015) obtains the non-asymptotic convergence rate of generalized conditional gradient method for convex composite optimization. Bach (2015) shows that the non-projected subgradient method for the primal convex composite optimization problem is equivalent to the conditional gradient applied to the dual optimization problem. Yu et al. (2017) improves generalized conditional gradient method for sparse optimization problems with convex gauge regularizers. Ghadimi (2019) focuses on smooth and weakly smooth nonconvex composite optimization problems. Ito et al. (2023) studies weakly convex composite optimization under Holder condition. Guo et al. (2022) provides a unified convergence analysis for zeroth-order conditional gradient methods on both stochastic constrained and composite optimization problems, under both convex and nonconvex settings. Recently, Chen et al. (2024); Assunção et al. (2025) extends conditional gradient methods to multiobjective composite optimization. See Braun et al. (2022) for a survey of conditional gradient methods.

## B    EXPERIMENTS ON REGULARIZED RESNET

We train a regularized Resnet-20 (He et al., 2016)[8] with cross-entropy loss for classification task on the Cifar 10 image data (Krizhevsky, 2009), using our 0-PGD algorithm (Algorithm 1) and 0-GCG algorithm (Algorithm 2). In particular, we use the objective function (18) where $\xi_i$ denotes an image-label pair in the Cifar 10 training set, and we select $\lambda_1 = \lambda_2 = 0.01$.

We implement our Algorithm 1 (0-PGD) and Algorithm 2 (0-GCG), and for each algorithm we test both gradient estimation options, G1 (minibatch) and G2 (variance reduction), all with radius $\delta = 0.001$. For both algorithms with option G1 we select batchsize 5000 and run 500 iterations.

---

[6]Proximal gradient method is called forward–backward splitting in (Attouch et al., 2013).

[7]Proximal gradient method is called iterative shrinkage-thresholding algorithms (ISTA) in (Beck & Teboulle, 2009)

[8]The Resnet-20 code comes from `https://github.com/sarwaridas/ResNet20_PyTorch/blob/main/resnet_cifar10_TRIAL.ipynb`

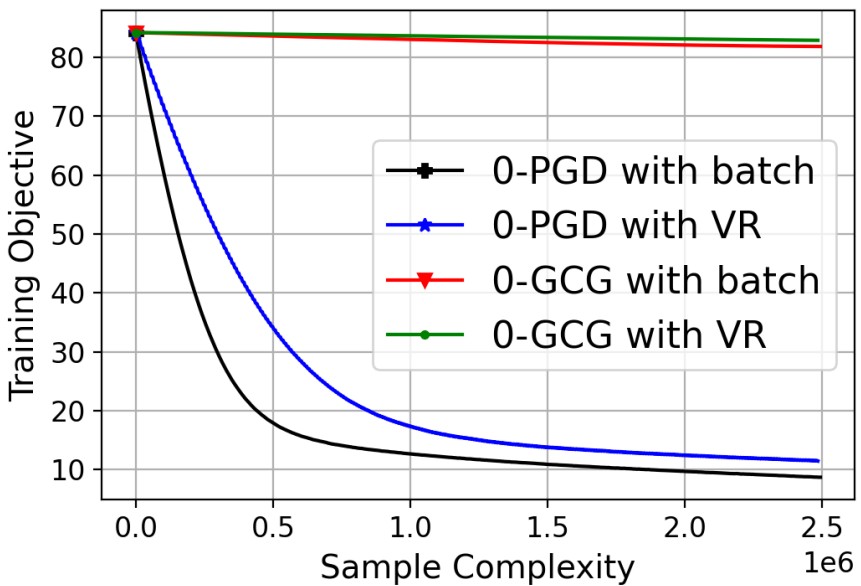

Figure 2: Experimental results on regularized Resnet-20.

For both algorithms with option G2 we run 2615 iterations, start each epoch of 10 iterations with batchsize 5000, and use batchsize 500 for the rest iterations. We use fine-tuned stepsizes 0.005 for 0-PGD with G1, 0.001 for 0-PGD with G2, $5 \times 10^{-5}$ for 0-GCG with G1, and $10^{-5}$ for 0-GCG with G2. The experiment is conducted on Python 3.11.5 in Red Hat Enterprise Linux 8.10 (Ootpa), using 1 RTX A6000 GPU (48GB memory) and 4 CPU cores (20GB memory).

At each iteration $t$, we evaluate the objective function $\phi(x_t)$ (Eq. (18)) on the Cifar-10 training data. In Figure 2, we plot $\phi(x_t)$ VS the function evaluation complexity (the total number of function evaluations up to each iteration $t$), which shows that Algorithm 1 (0-PGD) converges well while Algorithm 2 (0-GCG) descents on the objective very slowly. When tuning hyperparameters for 0-GCG, we found that 0-GCG ascents and diverges even with slightly larger stepsizes, and descents slightly faster with fine-tuned stepsizes when using larger batchsizes (e.g. 50000 for option G1 and start of each epoch of option G2, and 5000 for the rest iterations for option G2, which is time consuming). This phenomenon can be largely explained by comparing the theoretical batchsizes required by these algorithms, as shown in Table 2 below, which indicates that the batchsizes required by 0-GCG depend on the regularizer-dependent radius $R > 0$ defined by Assumption 3 while 0-PGD does not depend on $R$. Therefore, when $R$ is very large, 0-PGD can be much more efficient than 0-GCG.

Table 2: Batchsizes required by zeroth-order proximal gradient descent (0-PGD) and zeroth-order generalized conditional gradient algorithms (0-GCG).

| | 0-PGD (Algorithm 1) | 0-GCG (Algorithm 2) |
|---|---|---|
| Batchsize $B$ for option G1 | $\mathcal{O}(G^2\epsilon^{-2}d)$ (Theorem 2) | $\mathcal{O}(R^2G^2\epsilon^{-2}d)$ (Theorem 4) |
| Large batchsize $B_0$ for option G2 | $\mathcal{O}(G^2\epsilon^{-2}d)$ (Theorem 3) | $\mathcal{O}(R^2G^2\epsilon^{-2}d)$ (Theorem 5) |
| Small batchsize $B_1$ for option G2 | $\mathcal{O}(G\epsilon^{-1}\sqrt{d})$ (Theorem 3) | $\mathcal{O}(RG\epsilon^{-1}\sqrt{d})$ (Theorem 5) |

## C SUPPORTING LEMMAS

**Lemma 2** (Proposition 1 of (Ghadimi et al., 2016)). *For any $x, g, g' \in \mathbb{R}^d$, $\gamma > 0$ and proper convex function $h$, we have*

$$\|\mathcal{G}_{\gamma h}(x, g') - \mathcal{G}_{\gamma h}(x, g)\| \leq \|g' - g\| \tag{19}$$

**Lemma 3** (Lemma 1 of (Ghadimi et al., 2016)). *For any $x, g \in \mathbb{R}^d$, $\gamma > 0$ and proper convex function $h$, we have*

$$\langle g, \mathcal{G}_{\gamma h}(x, g) \rangle \geq \|\mathcal{G}_{\gamma h}(x, g)\|^2 + \frac{1}{\gamma}\big[h[\mathrm{prox}_{\gamma h}(x - \gamma g)] - h(x)\big]. \tag{20}$$

**Lemma 4.** *For i.i.d. random variables $\{X_i\}_{i=1}^N$, we have*

$$\mathbb{E}\Big[\Big\|\frac{1}{N}\sum_{i=1}^N[X_i - \mathbb{E}(X_i)]\Big\|^2\Big] = \frac{1}{N}\mathbb{E}\big[\|X_1 - \mathbb{E}X_1\|^2\big] \leq \frac{1}{N}\mathbb{E}\big[\|X_1\|^2\big] \tag{21}$$

*Proof.*

$$\mathbb{E}\Big[\Big\|\frac{1}{N}\sum_{i=1}^N[X_i - \mathbb{E}(X_i)]\Big\|^2\Big]$$

$$= \mathbb{E}\Big[\frac{1}{N^2}\sum_{i=1}^N\sum_{j=1}^N\big\langle X_i - \mathbb{E}(X_i), X_j - \mathbb{E}(X_j)\big\rangle\Big]$$

$$= \frac{1}{N^2}\sum_{i=1}^N\mathbb{E}[\|X_i - \mathbb{E}(X_i)\|^2] + \frac{1}{N^2}\sum_{i=1}^N\sum_{j=1,j\neq i}^N\mathbb{E}\big[\langle X_i - \mathbb{E}(X_i), X_j - \mathbb{E}(X_j)\rangle\big]$$

$$\overset{(a)}{=} \frac{1}{N^2}\sum_{i=1}^N\mathbb{E}[\|X_1 - \mathbb{E}(X_1)\|^2] + \frac{1}{N^2}\sum_{i=1}^N\sum_{j=1,j\neq i}^N\big[\langle\mathbb{E}[X_i - \mathbb{E}(X_i)], \mathbb{E}[X_j - \mathbb{E}(X_j)]\rangle\big]$$

$$\overset{(b)}{=} \frac{1}{N}\mathbb{E}[\|X_1 - \mathbb{E}(X_1)\|^2]$$

$$= \frac{1}{N}\mathbb{E}\big[\langle X_1 - \mathbb{E}(X_1), X_1 - \mathbb{E}(X_1)\rangle\big]$$

$$= \frac{1}{N}\mathbb{E}\big[\|X_1\|^2 - \langle\mathbb{E}(X_1), X_1\rangle - \langle X_1, \mathbb{E}(X_1)\rangle + \|\mathbb{E}(X_1)\|^2\big]$$

$$= \frac{1}{N}\mathbb{E}\big[\|X_1\|^2 - \|\mathbb{E}(X_1)\|^2\big]$$

$$\overset{(c)}{\leq} \frac{1}{N}\mathbb{E}\big[\|X_1\|^2\big] \tag{22}$$

where (a) uses the fact that $\{X_i\}_{i=1}^N$ are i.i.d. samples, (b) proves the "=" of Eq. (21), and (c) proves the "$\leq$" of Eq. (21). $\qquad\square$

**Lemma 5** (Lemma E.1 of (Lin et al., 2022)). *Suppose Assumption 1 holds and $\xi \sim \mathcal{P}$ and $u \in \mathrm{Uniform}(\mathcal{S}_d(1))$ (uniformly distribution in $\mathcal{S}_d(1) := \{y \in \mathbb{R}^d : \|y\| = 1\}$). Then for any $x \in \mathbb{R}^d$, the stochastic gradient estimator (4) satisfies $\mathbb{E}[\hat{g}_\delta(x; u, \xi)|x] = \nabla F_\delta(x)$ and $\mathbb{E}\big[\|\hat{g}_\delta(x; u, \xi)\|^2\big|x\big] \leq 16\sqrt{2\pi}dG^2$.*

**Lemma 6.** *Suppose Assumption 1 holds and we have i.i.d. samples $\{u_{i,t}\}_{i=1}^{B_t} \sim \mathrm{Uniform}(\mathcal{S}_d(1))$ and $\{\xi_{i,t}\}_{i=1}^{B_t} \sim \mathcal{P}$. Then the stochastic gradient estimator (9) satisfies $\mathbb{E}[g_t|x_t] = \nabla F_\delta(x_t)$ and $\mathbb{E}\big[\|g_t - \nabla F_\delta(x_t)\|^2\big|x_t\big] \leq \frac{16\sqrt{2\pi}dG^2}{B_t}$.*

*Proof.*

$$\mathbb{E}[g_t|x_t] = \mathbb{E}\Big[\frac{1}{B_t}\sum_{i=1}^{B_t}\hat{g}_\delta(x_t, u_{i,t}, \xi_{i,t})\Big|x_t\Big] = \frac{1}{B_t}\sum_{i=1}^{B_t}\mathbb{E}\big[\hat{g}_\delta(x_t, u_{i,t}, \xi_{i,t})|x_t\big] \overset{(a)}{=} \nabla F_\delta(x_t) \tag{23}$$

where (a) uses Lemma 5.

$$
\begin{aligned}
\mathbb{E}\big[\|g_t - \nabla F_\delta(x_t)\|^2\big|x_t\big] &= \mathbb{E}\Big[\Big\|\frac{1}{B_t}\sum_{i=1}^{B_t}[\hat{g}_\delta(x_t, u_{i,t}, \xi_{i,t}) - \nabla F_\delta(x_t)]\Big\|^2\Big|x_t\Big] \\
&\overset{(a)}{\leq} \frac{1}{B_t}\mathbb{E}\big[\|\hat{g}_\delta(x_t, u_{1,t}, \xi_{1,t})\|^2\big] \\
&\overset{(b)}{\leq} \frac{16\sqrt{2\pi}dG^2}{B_t}
\end{aligned}
\tag{24}
$$

where (a) uses $\mathbb{E}\big[\hat{g}_\delta(x_t, u_{i,t}, \xi_{i,t})\big|x_t\big] = \nabla F_\delta(x_t)$ (based on Lemma 5) and Lemma 4, and (b) uses Lemma 5. □

**Lemma 7** (Lemma A.5 of (Liu et al., 2024))**.** *Implement Algorithm 1 or 2 with Option G2. Select batchsize $B_t = B_0$ for any $t \bmod q = 0$ and $B_t = B_1$ for other $t$. Then under Assumption 1, the stochastic gradient estimation $g_t \approx \nabla F_\delta(x_t)$ has the following error bound.*

$$
\mathbb{E}\Big[\|g_t - \nabla F_\delta(x_t)\|^2\Big] \leq \frac{d^2 G^2}{\delta^2 B_1}\sum_{j=n_t q+1}^{t}\|x_j - x_{j-1}\|^2 + \frac{16\sqrt{2\pi}dG^2}{B_0}
\tag{25}
$$

*where $n_t = \lfloor t/q \rfloor$. Specifically, when $t \bmod q = 0$ (i.e., $t = n_t q$), the upper bound above reduces to $\frac{16\sqrt{2\pi}dG^2}{B_0}$.*

**Lemma 8.** *Implement Algorithm 2 with either Option G1 or G2, and the initialization $x_0$ satisfies $\|x_0 - x^{(h)}\| \leq R$. Then all the points $x_t, y_t$ generated from Algorithm 2 satisfies $\|x_t - x^{(h)}\| \leq R$ and $\|y_t - x^{(h)}\| \leq R$.*

*Proof.* Based on Proposition 3, $y_t \in \mathcal{L}_h(g_t)$ satisfies $\|y_t - x^{(h)}\| \leq R$.

We will prove $\|x_t - x^{(h)}\| \leq R$ by induction. Suppose $\|x_k - x^{(h)}\| \leq R$ for a certain natural number $k$. Then the update rule (15) implies that

$$
\|x_{k+1} - x^{(h)}\| = \|(1-\gamma)x_k + \gamma y_k\| \leq (1-\gamma)\|x_k\| + \gamma\|y_k\| \leq R.
$$

Since $\|x_0 - x^{(h)}\| \leq R$, we have proved that $\|x_t - x^{(h)}\| \leq R$ for all $t$. □

## D  PROOF OF PROPOSITION 1

Since $h$ is a proper and closed convex function based on Assumption 2, the sub-level set $A = \{x \in \mathbb{R}^d : \phi(x) \leq \phi(x^{(h)})\}$ is a closed set in which $h$ is continuous, based on Corollary 10.1.1 of (Rockafellar, 1970).

For any $x \in \mathbb{R}^d$ satisfying $\|x - x^{(h)}\| > R$, we have

$$
\phi(x) - \phi(x^{(h)}) \overset{(a)}{=} F(x) - F(x^{(h)}) + h(x) - h(x^{(h)}) \overset{(b)}{>} -G\|x - x^{(h)}\| + G\|x - x^{(h)}\| = 0,
$$

where (a) uses the objective function (1), (b) uses Assumption 3 and the $G$-Lipschitz continuity of $F$ (based on Assumption 1). Therefore, $A \subset \mathcal{B}_d(x^{(h)}, R) \overset{\text{def}}{=} \{x \in \mathbb{R}^d : \|x - x^{(h)}\| \leq R\}$, so $A$ is a compact set. Note that $\phi = F + h$ is continuous in $A$, so $\arg\max_{x \in A}\phi(x)$ is non-empty. Based on the definition of $A$, $\arg\max_{x \in \mathbb{R}^d} = \arg\max_{x \in A}\phi(x) \subset A \subset \mathcal{B}_d(x^{(h)}, R)$.

## E  PROOF OF PROPOSITION 2

**Proof of Item 1:** Item 1 is directly implied by Definition 5 and $\partial F(x) \subset \partial_\delta F(x)$.

**Proof of Item 2:** Since $x \in \mathbb{R}^d$ is a $(\gamma, \epsilon/(2L), \epsilon/2)$-PGSP, there exists $g \in \partial_\delta F(x) = \text{conv}\{\nabla F(y) : \|y - x\| \leq \frac{\epsilon}{2L}\}$ such that $\|\mathcal{G}_{\gamma h}(x, g)\| \leq \frac{\epsilon}{2}$. Therefore, $g$ can be written as the following convex combination of gradients

$$
g = \sum_{k=1}^{n}\alpha_k \nabla F(x_k),
$$

where $\alpha_k \geq 0$, $\sum_{k=1}^{n} \alpha_k = 1$ and $x_k \in \mathbb{R}^d$ satisfies $\|x_k - x\| \leq \frac{\epsilon}{2L}$. Then, we prove that $x$ is a $(\gamma, \epsilon)$-PGSP as follows.

$$\|\mathcal{G}_{\gamma h}[x, \nabla F(x)]\|$$

$$= \frac{1}{\gamma}\|x - \text{prox}_{\gamma h}[x - \gamma \nabla F(x)]\|$$

$$= \frac{1}{\gamma}\|x - \text{prox}_{\gamma h}(x - \gamma g)\| + \frac{1}{\gamma}\|\text{prox}_{\gamma h}(x - \gamma g) - \text{prox}_{\gamma h}[x - \gamma \nabla F(x)]\|$$

$$\overset{(a)}{\leq} \|\mathcal{G}_{\gamma h}(x, g)\| + \frac{1}{\gamma}\|(x - \gamma g) - [x - \gamma \nabla F(x)]\|$$

$$\leq \frac{\epsilon}{2} + \Big\|\sum_{k=1}^{n} \alpha_k [\nabla F(x_k) - \nabla F(x)]\Big\|$$

$$\leq \frac{\epsilon}{2} + \sum_{k=1}^{n} \alpha_k \|\nabla F(x_k) - \nabla F(x)\|$$

$$\overset{(b)}{\leq} \frac{\epsilon}{2} + L \sum_{k=1}^{n} \alpha_k \|x_k - x\|$$

$$\leq \frac{\epsilon}{2} + L \sum_{k=1}^{n} \alpha_k \Big(\frac{\epsilon}{2L}\Big) = \epsilon,$$

where (a) uses Lemma 2, and (b) uses the $L$-Lipschitz continuity of $\nabla F$.

**Proof of Item 3:** Item 3 is directly implied by Definition 5 and item 3 of Lemma 1.

## F    PROOF OF PROPOSITION 3

We will prove that Proposition 3 is a special case of Proposition 1. Specifically, in the original objective function (1), let $f_\xi(x) \equiv \langle\langle x, g \rangle$ that does not depend on the stochastic sample $\xi$, which is $L_\xi$-Lipschitz where $L_\xi \equiv G$ and thus satisfies Assumption 1. Therefore, by applying Proposition 3, we conclude that $\arg\min_{y \in \mathbb{R}^d}[h(y) + \langle y, g \rangle] = \arg\min_{y \in \mathbb{R}^d} \phi(x)$ is a non-empty subset of $\mathcal{B}_d(x^{(h)}, R)$.

## G    PROOF OF PROPOSITION 4

**Proof of Item 1:** Item 1 is directly implied by Definition 7 and $\partial F(x) \subset \partial_\delta F(x)$.

**Proof of Item 2:** If $x \in \mathbb{R}^d$ is an $(\epsilon/(2RL), \epsilon/2)$-CGGSP, then based on Definition 7, there exists $g \in \partial_\delta F(x)$ such that

$$\max_{y \in \mathbb{R}^d} \big[h(x) - h(y) + \langle y - x, -g \rangle\big] \leq \frac{\epsilon}{2}. \tag{26}$$

As $F$ is differentiable, $g$ can be written as the following convex combination of gradients

$$g = \sum_{k=1}^{n} \alpha_k \nabla F(x_k),$$

where $\alpha_k \geq 0$, $\sum_{k=1}^{n} \alpha_k = 1$ and $x_k \in \mathbb{R}^d$ satisfies $\|x_k - x\| \leq \frac{\epsilon}{2RL}$. Then we can prove Item 2 as follows.

$$\mathcal{W}_h[x, \nabla F(x)] \overset{(a)}{=} h(x) - h(y) + \langle y - x, -\nabla F(x) \rangle$$

$$= h(x) - h(y) + \langle y - x, -g \rangle + \langle y - x, g - \nabla F(x) \rangle$$

$$\leq h(x) - h(y) + \langle y - x, -g \rangle + \|y - x\| \cdot \|g - \nabla F(x)\|$$

$$\overset{(b)}{\leq} \frac{\epsilon}{2} + R\Big\|\sum_{k=1}^{n} \alpha_k [\nabla F(x_k) - \nabla F(x)]\Big\|$$

$$\leq \frac{\epsilon}{2} + R\sum_{k=1}^{n} \alpha_k \|\nabla F(x_k) - \nabla F(x)\|$$

$$\overset{(c)}{\leq} \frac{\epsilon}{2} + R\sum_{k=1}^{n} \alpha_k L \|x_k - x\|$$

$$\overset{(d)}{\leq} \frac{\epsilon}{2} + \frac{\epsilon}{2} = \epsilon,$$

where (a) holds for any $y \in \mathcal{L}_h[x, \nabla F(x)]$ based on Eqs. (8) and (7), (b) uses Eq. (26), Proposition 3 and $\sum_{k=1}^{n} \alpha_k = 1$, (c) uses the $L$-Lipschitz continuity of $\nabla F$, and (d) uses $\|x_k - x\| \leq \frac{\epsilon}{2RL}$ and $\sum_{k=1}^{n} \alpha_k = 1$.

**Proof of Item 3:** Item 3 is directly implied by Definition 7 and item 3 of Lemma 1.

## H  PROOF OF THEOREM 1

The proof of Theorem 4.7 in (Liu et al., 2024) has designed a function $F$[9] that satisfies Assumption 1 that $F$ is Lipschitz continuous, and satisfies $F(0) - \inf_{x \in \Omega} F(x) \leq 2$ where $\Omega = [-100, 100]^d$ is a convex and compact set. The conclusion of their Theorem 4.7 states each point $x_t$ generated from the algorithm $\mathcal{A}$ satisfies the following inequalities for $\gamma \in (0, 0.1]$.

$$\min_{g \in \partial F(x_t)} \left[ \frac{1}{\gamma} \|x_t - \psi(x_t, g, \gamma)\| \right] \geq \frac{1}{4\sqrt{2}}, \tag{27}$$

$$\min_{g \in \partial F(x_t)} \max_{u \in \Omega} \langle u - x_t, -g \rangle \geq \frac{1}{4\sqrt{2}}, \tag{28}$$

where

$$\psi(x, g, \gamma) = \arg\min_{y \in \Omega} \left( \langle g, y \rangle + \frac{1}{2\gamma} \|y - x\|^2 \right). \tag{29}$$

Select $h = h_\Omega$ defined by Eq. (2), which yields the following equations.

$$\text{prox}_{\gamma h}(x_t - \gamma g) \overset{(a)}{=} \arg\min_{y \in \mathbb{R}^n} \left[ h_\Omega(y) + \frac{1}{2\gamma} \|y - x_t + \gamma g\|^2 \right]$$

$$\overset{(b)}{=} \arg\min_{y \in \Omega} \left[ \frac{1}{2\gamma} \|y - x_t\|^2 + \langle g, y - x_t \rangle \right] \overset{(c)}{=} \psi(x_t, g, \gamma), \tag{30}$$

$$\min_{g \in \partial F(x_t)} \mathcal{W}_h(x_t, g) \overset{(d)}{=} \min_{g \in \partial F(x_t)} \max_{y \in \mathbb{R}^d} \left[ h_\Omega(x_t) - h_\Omega(y) + \langle y - x_t, -g \rangle \right]$$

$$\overset{(b)}{\geq} \min_{g \in \partial F(x_t)} \max_{y \in \Omega} \langle y - x_t, -g \rangle \overset{(e)}{\geq} \frac{1}{4\sqrt{2}}, \tag{31}$$

$$\phi(0) - \inf_{x \in \mathbb{R}^d} \phi(x) = F(0) + h_\Omega(0) - \inf_{x \in \mathbb{R}^d} [F(x) + h_\Omega(x)] \overset{(b)}{=} F(0) - \inf_{x \in \Omega} F(x) \leq 2, \tag{32}$$

where (a)-(e) use Eqs. (5), (2), (29), (8) and (28) respectively. Therefore,

$$\min_{g \in \partial F(x_t)} \|\mathcal{G}_{\gamma h}(x_t, g)\| \overset{(a)}{=} \min_{g \in \partial F(x_t)} \left[ \frac{1}{\gamma} \|x_t - \text{prox}_{\gamma h}(x_t - \gamma g)\| \right]$$

$$\overset{(b)}{=} \min_{g \in \partial F(x_t)} \left[ \frac{1}{\gamma} \|x_t - \psi(x_t, g, \gamma)\| \right] \overset{(c)}{\geq} \frac{1}{4\sqrt{2}}, \tag{33}$$

where (a) uses Eq. (6), (b) uses Eq. (30) and (c) uses Eq. (27).

Eq. (33) implies that $x_t$ is not a $(\gamma, \epsilon)$-PGSP for $\epsilon \leq \frac{1}{4\sqrt{2}}$. Eq. (31) implies that $x_t$ is not an $\epsilon$-CGGSP for $\epsilon \leq \frac{1}{4\sqrt{2}}$. These implications along with Eq. (32) conclude the proof.

---

[9]This function is denoted as $F_w$ in (Liu et al., 2024).

## I   PROOF OF THEOREM 2

Since $\nabla F_\delta$ is $\frac{cG\sqrt{d}}{\delta}$-Lipschitz continuous based on item 2 of Lemma 1, we obtain that

$$
\begin{aligned}
F_\delta(x_{t+1}) \leq & F_\delta(x_t) + \langle \nabla F_\delta(x_t), x_{t+1} - x_t \rangle + \frac{cG\sqrt{d}}{2\delta}\|x_{t+1} - x_t\|^2 \\
\overset{(a)}{=} & F_\delta(x_t) - \gamma\langle \nabla F_\delta(x_t), \mathcal{G}_{\gamma h}(x_t, g_t)\rangle + \frac{cG\gamma^2\sqrt{d}}{2\delta}\|\mathcal{G}_{\gamma h}(x_t, g_t)\|^2 \\
= & F_\delta(x_t) - \gamma\langle g_t, \mathcal{G}_{\gamma h}(x_t, g_t)\rangle + \gamma\langle g_t - \nabla F_\delta(x_t), \mathcal{G}_{\gamma h}(x_t, g_t)\rangle + \frac{cG\gamma^2\sqrt{d}}{2\delta}\|\mathcal{G}_{\gamma h}(x_t, g_t)\|^2 \\
\overset{(b)}{\leq} & F_\delta(x_t) - \gamma\Big[\|\mathcal{G}_{\gamma h}(x_t, g_t)\|^2 + \frac{1}{\gamma}\big[h[\mathrm{prox}_{\gamma h}(x_t - \gamma g_t)] - h(x_t)\big]\Big] \\
& + \gamma\langle g_t - \nabla F_\delta(x_t), \mathcal{G}_{\gamma h}(x_t, g_t)\rangle + \frac{\gamma}{2}\|\mathcal{G}_{\gamma h}(x_t, g_t)\|^2 \\
\overset{(c)}{=} & F_\delta(x_t) - \frac{\gamma}{2}\|\mathcal{G}_{\gamma h}(x_t, g_t)\|^2 + h(x_t) - h(x_{t+1}) + \gamma\langle g_t - \nabla F_\delta(x_t), \mathcal{G}_{\gamma h}[x_t, \nabla F_\delta(x_t)]\rangle \\
& + \gamma\langle g_t - \nabla F_\delta(x_t), \mathcal{G}_{\gamma h}(x_t, g_t) - \mathcal{G}_{\gamma h}[x_t, \nabla F_\delta(x_t)]\rangle \\
\leq & F_\delta(x_t) - \frac{\gamma}{2}\|\mathcal{G}_{\gamma h}(x_t, g_t)\|^2 + h(x_t) - h(x_{t+1}) + \gamma\langle g_t - \nabla F_\delta(x_t), \mathcal{G}_{\gamma h}[x_t, \nabla F_\delta(x_t)]\rangle \\
& + \gamma\|g_t - \nabla F_\delta(x_t)\| \cdot \big\|\mathcal{G}_{\gamma h}(x_t, g_t) - \mathcal{G}_{\gamma h}[x_t, \nabla F_\delta(x_t)]\big\| \\
\overset{(d)}{\leq} & F_\delta(x_t) - \frac{\gamma}{2}\|\mathcal{G}_{\gamma h}(x_t, g_t)\|^2 + h(x_t) - h(x_{t+1}) + \gamma\langle g_t - \nabla F_\delta(x_t), \mathcal{G}_{\gamma h}[x_t, \nabla F_\delta(x_t)]\rangle \\
& + \gamma\|g_t - \nabla F_\delta(x_t)\|^2,
\end{aligned}
$$

where (a) and (c) use the update rule (11) that $x_{t+1} = \mathrm{prox}_{\gamma h}(x_t - \gamma g_t) = x_t - \gamma\mathcal{G}_{\gamma h}(x_t, g_t)$, (b) uses Lemma 3 and the stepsize $\gamma = \frac{\delta}{cG\sqrt{d}}$, and (d) uses Lemma 2. Rearranging the inequality above, and taking expectation, we obtain that

$$
\begin{aligned}
\frac{\gamma}{2}\mathbb{E}[\|\mathcal{G}_{\gamma h}(x_t, g_t)\|^2] \leq & \mathbb{E}[F_\delta(x_t) + h(x_t) - F_\delta(x_{t+1}) - h(x_{t+1})] \\
& + \gamma\mathbb{E}\big[\langle g_t - \nabla F_\delta(x_t), \mathcal{G}_{\gamma h}[x_t, \nabla F_\delta(x_t)]\rangle\big] + \gamma\mathbb{E}[\|g_t - \nabla F_\delta(x_t)\|^2] \\
\leq & \mathbb{E}[F_\delta(x_t) + h(x_t) - F_\delta(x_{t+1}) - h(x_{t+1})] + \frac{16\gamma\sqrt{2\pi}dG^2}{B_t},
\end{aligned}
$$

where the second $\leq$ uses Lemma 6. Rearranging the inequality above and summing over $t = 0, 1, \ldots, T - 1$, we have

$$
\begin{aligned}
\mathbb{E}[\|\mathcal{G}_{\gamma h}(x_{\widetilde{T}}, g_{\widetilde{T}})\|^2] = & \frac{1}{T}\sum_{t=0}^{T-1}\mathbb{E}[\|\mathcal{G}_{\gamma h}(x_t, g_t)\|^2] \\
\leq & \frac{1}{T}\sum_{t=0}^{T-1}\Big[\frac{2}{\gamma}\mathbb{E}[F_\delta(x_t) + h(x_t) - F_\delta(x_{t+1}) - h(x_{t+1})] + \frac{32\sqrt{2\pi}dG^2}{B_t}\Big] \\
\overset{(a)}{\leq} & \frac{2}{T\gamma}\mathbb{E}[F_\delta(x_0) + h(x_0) - F_\delta(x_T) - h(x_T)] + \frac{32\sqrt{2\pi}dG^2}{B} \\
\overset{(b)}{\leq} & \frac{2cG\sqrt{d}}{T\delta}\mathbb{E}[F(x_0) + h(x_0) - F(x_T) - h(x_T) + 2\delta G] + \frac{32\sqrt{2\pi}dG^2}{B} \\
\overset{(c)}{\leq} & \frac{2cG\sqrt{d}}{T\delta}\mathbb{E}[\phi(x_0) - \phi_{\min} + 2\delta G] + \frac{32\sqrt{2\pi}dG^2}{B}, \quad\quad (34)
\end{aligned}
$$

where (a) uses constant batchsize $B_t \equiv B$ and stepsize $\gamma = \frac{\delta}{cG\sqrt{d}}$, (b) uses item 1 of Lemma 1, (c) uses $\phi \overset{\text{def}}{=} F + g$ and $\phi_{\min} \overset{\text{def}}{=} \min_{x\in\mathbb{R}^d}\phi(x)$. Therefore, we can obtain the convergence rate (12) as follows.

$$
\mathbb{E}[\|\mathcal{G}_{\gamma h}(x_{\widetilde{T}}, \nabla F_\delta(x_{\widetilde{T}}))\|] \leq \mathbb{E}[\|\mathcal{G}_{\gamma h}(x_{\widetilde{T}}, g_{\widetilde{T}})\|] + \mathbb{E}[\|\mathcal{G}_{\gamma h}(x_{\widetilde{T}}, \nabla F_\delta(x_{\widetilde{T}})) - \mathcal{G}_{\gamma h}(x_{\widetilde{T}}, g_{\widetilde{T}})\|]
$$

$$\overset{(a)}{\leq} \sqrt{\mathbb{E}[\|\mathcal{G}_{\gamma h}(x_{\widetilde{T}}, g_{\widetilde{T}})\|^2]} + \sqrt{\mathbb{E}[\|\nabla F_\delta(x_{\widetilde{T}}) - g_{\widetilde{T}}\|^2]}$$

$$\overset{(b)}{\leq} \sqrt{\frac{2cG\sqrt{d}}{T\delta} \mathbb{E}[\phi(x_0) - \phi_{\min} + 2\delta G] + \frac{32\sqrt{2\pi}dG^2}{B}} + \sqrt{\frac{16\sqrt{2\pi}dG^2}{B}}$$

$$\overset{(c)}{\leq} \frac{\sqrt{2cG}d^{1/4}}{\sqrt{T\delta}} \sqrt{\mathbb{E}[\phi(x_0)] - \phi_{\min} + 2\delta G} + \frac{16G\sqrt{d}}{\sqrt{B}},$$

where (a) uses Lemma 2, (b) uses Eq. (34) and Lemma 6, (c) uses $\sqrt{a+b} \leq \sqrt{a} + \sqrt{b}$ for any $a, b \geq 0$ and $\sqrt{32\sqrt{2\pi}} + \sqrt{16\sqrt{2\pi}} < 16$

Furthermore, we can select the following hyperparameters.

$$T = \frac{8cG\sqrt{d}}{\delta\epsilon^2} \big[\mathbb{E}[\phi(x_0)] - \phi_{\min} + 2\delta G\big] = \mathcal{O}(Gd^{1/2}\delta^{-1}\epsilon^{-2}), \tag{35}$$

$$B = \frac{1024dG^2}{\epsilon^2} = \mathcal{O}(G^2 d\epsilon^{-2}). \tag{36}$$

Then substituting the hyperparameters above into the convergence rate (12), we obtain the following bound, which based on item 3 of Proposition 2 implies that there exists at least one $(\gamma, \delta, \epsilon)$-PGSP in $\{x_t\}_{t=0}^{T-1}$.

$$\min_{0 \leq t \leq T-1} \mathbb{E}[\|\mathcal{G}_{\gamma h}(x_t, \nabla F_\delta(x_t))\|] \leq \mathbb{E}[\|\mathcal{G}_{\gamma h}(x_{\widetilde{T}}, \nabla F_\delta(x_{\widetilde{T}}))\|] \leq \epsilon.$$

## J  PROOF OF THEOREM 3

Since $\nabla F_\delta$ is $\frac{cG\sqrt{d}}{\delta}$-Lipschitz continuous based on item 2 of Lemma 1, we obtain that

$$F_\delta(x_{t+1}) \leq F_\delta(x_t) + \langle \nabla F_\delta(x_t), x_{t+1} - x_t \rangle + \frac{cG\sqrt{d}}{2\delta} \|x_{t+1} - x_t\|^2$$

$$\overset{(a)}{=} F_\delta(x_t) - \gamma \langle \nabla F_\delta(x_t), \mathcal{G}_{\gamma h}(x_t, g_t) \rangle + \frac{cG\gamma^2\sqrt{d}}{2\delta} \|\mathcal{G}_{\gamma h}(x_t, g_t)\|^2$$

$$= F_\delta(x_t) - \gamma \langle g_t, \mathcal{G}_{\gamma h}(x_t, g_t) \rangle + \gamma \langle g_t - \nabla F_\delta(x_t), \mathcal{G}_{\gamma h}(x_t, g_t) \rangle + \frac{cG\gamma^2\sqrt{d}}{2\delta} \|\mathcal{G}_{\gamma h}(x_t, g_t)\|^2$$

$$\overset{(b)}{\leq} F_\delta(x_t) - \gamma \Big[ \|\mathcal{G}_{\gamma h}(x_t, g_t)\|^2 + \frac{1}{\gamma} \big[h[\text{prox}_{\gamma h}(x_t - \gamma g_t)] - h(x_t)\big] \Big]$$

$$+ \frac{\gamma}{2} \|g_t - \nabla F_\delta(x_t)\|^2 + \Big( \frac{cG\gamma^2\sqrt{d}}{2\delta} + \frac{\gamma}{2} \Big) \|\mathcal{G}_{\gamma h}(x_t, g_t)\|^2$$

$$\overset{(c)}{=} F_\delta(x_t) + h(x_t) - h(x_{t+1}) + \frac{\gamma}{2} \|g_t - \nabla F_\delta(x_t)\|^2 + \Big( \frac{cG\gamma^2\sqrt{d}}{2\delta} - \frac{\gamma}{2} \Big) \|\mathcal{G}_{\gamma h}(x_t, g_t)\|^2,$$

where (a) and (c) use the update rule (11) that $x_{t+1} = \text{prox}_{\gamma h}(x_t - \gamma g_t) = x_t - \gamma \mathcal{G}_{\gamma h}(x_t, g_t)$, and (b) uses Lemma 3 and the inequality that $\langle u, v \rangle \leq (\|u\|^2 + \|v\|^2)/2$ for $u = g_t - \nabla F_\delta(x_t)$, $v = \mathcal{G}_{\gamma h}(x_t, g_t)$. Rearranging the inequality above, and taking expectation, we obtain that

$$\Big( \frac{\gamma}{2} - \frac{cG\gamma^2\sqrt{d}}{2\delta} \Big) \mathbb{E}[\|\mathcal{G}_{\gamma h}(x_t, g_t)\|^2]$$

$$\leq \mathbb{E}[F_\delta(x_t) + h(x_t) - F_\delta(x_{t+1}) - h(x_{t+1})] + \frac{\gamma}{2} \mathbb{E}[\|g_t - \nabla F_\delta(x_t)\|^2]$$

$$\overset{(a)}{\leq} \mathbb{E}[F_\delta(x_t) + h(x_t) - F_\delta(x_{t+1}) - h(x_{t+1})] + \frac{\gamma d^2 G^2}{2\delta^2 B_1} \sum_{j=n_t q+1}^{t} \mathbb{E}[\|x_j - x_{j-1}\|^2] + \frac{8\gamma\sqrt{2\pi}dG^2}{B_0},$$

where (a) uses Lemma 7. Summing the inequality above over $t = 0, 1, \ldots, T-1$, we have

$$\Big( \frac{\gamma}{2} - \frac{cG\gamma^2\sqrt{d}}{2\delta} \Big) \sum_{t=0}^{T-1} \mathbb{E}[\|\mathcal{G}_{\gamma h}(x_t, g_t)\|^2]$$

$$\leq \mathbb{E}[F_\delta(x_0) + h(x_0) - F_\delta(x_T) - h(x_T)] + \frac{\gamma d^2 G^2}{2\delta^2 B_1} \sum_{t=0}^{T-1} \sum_{j=n_t q+1}^{t} \|x_j - x_{j-1}\|^2 + \frac{8T\gamma\sqrt{2\pi}dG^2}{B_0}$$

$$\overset{(a)}{\leq} \mathbb{E}[F(x_0) + h(x_0) - F(x_T) - h(x_T)] + 2\delta G$$

$$+ \frac{\gamma^3 d^2 G^2}{2\delta^2 B_1} \sum_{t=0}^{T-1} \sum_{j=n_t q+1}^{(n_t+1)q} \|\mathcal{G}_{\gamma h}(x_j, g_j)\|^2 + \frac{8T\gamma\sqrt{2\pi}dG^2}{B_0}$$

$$\overset{(b)}{\leq} \mathbb{E}[\phi(x_0) - \phi_{\min}] + 2\delta G + \frac{q\gamma^3 d^2 G^2}{2\delta^2 B_1} \sum_{t=0}^{T-1} \mathbb{E}[\|\mathcal{G}_{\gamma h}(x_t, g_t)\|^2] + \frac{8T\gamma\sqrt{2\pi}dG^2}{B_0}, \tag{37}$$

where (a) uses $t < (n_t + 1)q$ ($n_t = \lfloor t/q \rfloor$), item 1 of Lemma 1 and the update rule (11) that $x_{j+1} = \text{prox}_{\gamma h}(x_j - \gamma g_j) = x_j - \gamma\mathcal{G}_{\gamma h}(x_j, g_j)$, and (b) uses $\phi \overset{\text{def}}{=} F + g$ and $\phi_{\min} \overset{\text{def}}{=} \min_{x\in\mathbb{R}^d} \phi(x)$. Rearranging the inequality above, we obtain that

$$\mathbb{E}[\|\mathcal{G}_{\gamma h}(x_{\widetilde{T}}, g_{\widetilde{T}})\|^2]$$

$$= \frac{1}{T} \sum_{t=0}^{T-1} \mathbb{E}[\|\mathcal{G}_{\gamma h}(x_t, g_t)\|^2]$$

$$\leq \frac{1}{T}\Big(\frac{\gamma}{2} - \frac{cG\gamma^2\sqrt{d}}{2\delta} - \frac{q\gamma^3 d^2 G^2}{2\delta^2 B_1}\Big)^{-1}\Big[\mathbb{E}[\phi(x_0)] - \phi_{\min} + 2\delta G + \frac{8T\gamma\sqrt{2\pi}dG^2}{B_0}\Big]$$

$$\leq \frac{16G(d + c\sqrt{d})}{T\delta}\Big[\mathbb{E}[\phi(x_0)] - \phi_{\min} + 2\delta G\Big] + \frac{64\sqrt{2\pi}dG^2}{B_0} \tag{38}$$

where (a) uses the following inequality with stepsize $\gamma = \frac{\delta}{2G(d+c\sqrt{d})}$ and batchsize $B_1 = q$, and (b) uses stepsize $\gamma = \frac{\delta}{2G(d+c\sqrt{d})}$.

$$\frac{\gamma}{2} - \frac{cG\gamma^2\sqrt{d}}{2\delta} - \frac{q\gamma^3 d^2 G^2}{2\delta^2 B_1} = \frac{\gamma}{2}\Big(1 - \frac{cG\gamma\sqrt{d}}{\delta} - \frac{\gamma^2 d^2 G^2}{\delta^2}\Big)$$

$$\geq \frac{\delta}{4G(d + c\sqrt{d})}\Big(1 - \frac{c\sqrt{d}}{2(d + c\sqrt{d})} - \frac{d^2}{4(d + c\sqrt{d})^2}\Big)$$

$$\geq \frac{\delta}{4G(d + c\sqrt{d})}\Big(1 - \frac{1}{2} - \frac{1}{4}\Big) = \frac{\delta}{16G(d + c\sqrt{d})}.$$

Then we have the following bound.

$$\mathbb{E}[\|\mathcal{G}_{\gamma h}(x_{\widetilde{T}}, \nabla F_\delta(x_{\widetilde{T}}))\|^2]$$

$$\leq 2\mathbb{E}[\|\mathcal{G}_{\gamma h}(x_{\widetilde{T}}, g_{\widetilde{T}})\|^2] + 2\mathbb{E}[\|\mathcal{G}_{\gamma h}(x_{\widetilde{T}}, \nabla F_\delta(x_{\widetilde{T}})) - \mathcal{G}_{\gamma h}(x_{\widetilde{T}}, g_{\widetilde{T}})\|^2]$$

$$\overset{(a)}{\leq} 2\mathbb{E}[\|\mathcal{G}_{\gamma h}(x_{\widetilde{T}}, g_{\widetilde{T}})\|^2] + 2\mathbb{E}[\|\nabla F_\delta(x_{\widetilde{T}}) - g_{\widetilde{T}}\|^2]$$

$$= \frac{2}{T} \sum_{t=0}^{T-1} \Big[\mathbb{E}[\|\mathcal{G}_{\gamma h}(x_t, g_t)\|^2] + \mathbb{E}[\|\nabla F_\delta(x_t) - g_t\|^2]\Big]$$

$$\overset{(b)}{\leq} \frac{2}{T} \sum_{t=0}^{T-1} \Big[\mathbb{E}[\|\mathcal{G}_{\gamma h}(x_t, g_t)\|^2] + \frac{d^2 G^2}{\delta^2 B_1} \sum_{j=n_t q+1}^{t} \|x_j - x_{j-1}\|^2 + \frac{16\sqrt{2\pi}dG^2}{B_0}\Big]$$

$$\overset{(c)}{\leq} \frac{2}{T} \sum_{t=0}^{T-1} \mathbb{E}[\|\mathcal{G}_{\gamma h}(x_t, g_t)\|^2] + \frac{2d^2 G^2 \gamma^2}{T\delta^2 q} \sum_{t=0}^{T-1} \sum_{j=n_t q+1}^{(n_t+1)q} \mathbb{E}[\|\mathcal{G}_{\gamma h}(x_j, g_j)\|^2]$$

$$\leq \Big(\frac{2}{T} + \frac{2d^2 G^2 \gamma^2}{T\delta^2}\Big) \sum_{t=0}^{T-1} \mathbb{E}[\|\mathcal{G}_{\gamma h}(x_t, g_t)\|^2]$$

$$\overset{(d)}{\leq} \Big(2 + \frac{d^2}{2(d + c\sqrt{d})^2}\Big)\Big\{\frac{16G(d + c\sqrt{d})}{T\delta}\Big[\mathbb{E}[\phi(x_0)] - \phi_{\min} + 2\delta G\Big] + \frac{64\sqrt{2\pi}dG^2}{B_0}\Big\}$$

$$\leq \frac{40G(d + c\sqrt{d})}{T\delta}\Big[\mathbb{E}[\phi(x_0)] - \phi_{\min} + 2\delta G\Big] + \frac{160\sqrt{2\pi}dG^2}{B_0},$$

where (a) uses Lemma 2, (b) uses Lemma 7, (c) uses $t < (n_t + 1)q$, $B_1 = q$ and the update rule that $x_j = x_{j-1} + \gamma\mathcal{G}_{\gamma h}(x_j, g_j)$, and (d) uses Eq. (38) and $\gamma = \frac{\delta}{2G(d + c\sqrt{d})}$. Therefore, by taking square root of the inequality above, we obtain the convergence rate (13) as follows.

$$\mathbb{E}[\|\mathcal{G}_{\gamma h}(x_{\widetilde{T}}, \nabla F_\delta(x_{\widetilde{T}}))\|] \leq \sqrt{\mathbb{E}[\|\mathcal{G}_{\gamma h}(x_{\widetilde{T}}, \nabla F_\delta(x_{\widetilde{T}}))\|^2]}$$

$$\leq \sqrt{\frac{40G(d + c\sqrt{d})}{T\delta}\Big[\mathbb{E}[\phi(x_0)] - \phi_{\min} + 2\delta G\Big] + \frac{160\sqrt{2\pi}dG^2}{B_0}}$$

$$\leq \frac{\sqrt{40G}(\sqrt{d} + \sqrt{c}d^{1/4})}{\sqrt{T\delta}}\sqrt{\mathbb{E}[\phi(x_0)] - \phi_{\min} + 2\delta G} + \frac{21G\sqrt{d}}{\sqrt{B_0}},$$

where the final $\leq$ uses $\sqrt{160\sqrt{2\pi}} < 21$ and $\sqrt{a + b} \leq \sqrt{a} + \sqrt{b}$ for any $a, b \geq 0$.

Furthermore, we can select $B_0 = 1764dG^2\epsilon^{-2}$, $B_1 = q = 42\sqrt{d}G\epsilon^{-1}$ and the following $T$

$$T = 320\delta^{-1}\epsilon^{-2}G(d + c\sqrt{d})\big[\mathbb{E}[\phi(x_0)] - \phi_{\min} + 2\delta G\big] = \mathcal{O}(Gd\delta^{-1}\epsilon^{-2}). \quad (39)$$

Then substituting the hyperparameters above into the convergence rate (13), we obtain the following bound, which based on item 3 of Proposition 2 implies that there exists at least one $(\gamma, \delta, \epsilon)$-PGSP in $\{x_t\}_{t=0}^{T-1}$.

$$\min_{0 \leq t \leq T-1} \mathbb{E}[\|\mathcal{G}_{\gamma h}(x_t, \nabla F_\delta(x_t))\|] \leq \mathbb{E}[\|\mathcal{G}_{\gamma h}(x_{\widetilde{T}}, \nabla F_\delta(x_{\widetilde{T}}))\|] \leq \epsilon.$$

## K  PROOF OF THEOREM 4

Denote $\widetilde{y}_t \in \mathcal{L}_h[\nabla F_\delta(x_t)] \overset{\text{def}}{=} \arg\min_{y \in \mathbb{R}^d}[h(y) + \langle y, \nabla F_\delta(x_t)\rangle]$. Then the $\delta$-regularized Frank-Wolfe gap (8) can be expressed as follows.

$$\mathcal{W}_h[x_t, \nabla F_\delta(x_t)] \overset{\text{def}}{=} \max_{y \in \mathbb{R}^d}\big[h(x_t) - h(y) + \langle y - x_t, -\nabla F_\delta(x_t)\rangle\big]$$

$$= h(x_t) - h(\widetilde{y}_t) - \langle \widetilde{y}_t - x_t, \nabla F_\delta(x_t)\rangle. \quad (40)$$

Then we have

$$\langle\nabla F_\delta(x_t), y_t - x_t\rangle = \langle\nabla F_\delta(x_t), \widetilde{y}_t - x_t\rangle + \langle\nabla F_\delta(x_t), y_t - \widetilde{y}_t\rangle$$

$$= \langle\nabla F_\delta(x_t), \widetilde{y}_t - x_t\rangle + \langle g_t, y_t - \widetilde{y}_t\rangle + \langle\nabla F_\delta(x_t) - g_t, y_t - \widetilde{y}_t\rangle$$

$$\overset{(a)}{\leq} \langle\nabla F_\delta(x_t), \widetilde{y}_t - x_t\rangle + h(\widetilde{y}_t) - h(y_t) + \|\nabla F_\delta(x_t) - g_t\|\|y_t - \widetilde{y}_t\|$$

$$\overset{(b)}{\leq} -\mathcal{W}_h[x_t, \nabla F_\delta(x_t)] + h(x_t) - h(y_t) + 2R\|\nabla F_\delta(x_t) - g_t\| \quad (41)$$

where (a) uses $h(y_t) + \langle y_t, g_t\rangle \leq h(\widetilde{y}_t) + \langle\widetilde{y}_t, g_t\rangle$ based on the update rule (14), and (b) uses Eq. (40) as well as $\|y_t - x^{(h)}\| \leq R$ and $\|\widetilde{y}_t - x^{(h)}\| \leq R$ (based on Proposition 3 and item 2 of Lemma 1).

Then since $\nabla F_\delta$ is $\frac{cG\sqrt{d}}{\delta}$-Lipschitz continuous based on item 2 of Lemma 1, we obtain that

$$F_\delta(x_{t+1})$$

$$\leq F_\delta(x_t) + \langle\nabla F_\delta(x_t), x_{t+1} - x_t\rangle + \frac{cG\sqrt{d}}{2\delta}\|x_{t+1} - x_t\|^2$$

$$\overset{(a)}{=} F_\delta(x_t) + \gamma\langle\nabla F_\delta(x_t), y_t - x_t\rangle + \frac{cG\sqrt{d}\gamma^2}{2\delta}\|y_t - x_t\|^2$$

$$\overset{(b)}{\leq} F_\delta(x_t) - \gamma\mathcal{W}_h[x_t, \nabla F_\delta(x_t)] + \gamma h(x_t) - \gamma h(y_t) + 2R\gamma\|\nabla F_\delta(x_t) - g_t\| + \frac{cG\sqrt{d}(2R)^2\gamma^2}{2\delta}$$

$$\overset{(c)}{\leq} F_\delta(x_t) - \gamma\mathcal{W}_h[x_t, \nabla F_\delta(x_t)] + h(x_t) - h(x_{t+1}) + 2R\gamma\|\nabla F_\delta(x_t) - g_t\| + \frac{2cG\sqrt{d}R^2\gamma^2}{\delta},$$

where (a) uses the update rule (15), (b) uses Eq. (41) and Lemma 8, and (c) uses $h(x_{t+1}) \leq (1-\gamma)h(x_t) + \gamma h(y_t)$ which holds for convex function $h$ and $x_{t+1}$ obtained from the update rule (15). Rearranging the inequality above and averaging it over $t = 0, 1, \ldots, T-1$, we obtain the convergence rate (16) as follows.

$$
\mathbb{E}\big[\mathcal{W}_h[x_{\widetilde{T}}, \nabla F_\delta(x_{\widetilde{T}})]\big]
$$

$$
= \frac{1}{T} \sum_{t=0}^{T-1} \mathcal{W}_h[x_t, \nabla F_\delta(x_t)]
$$

$$
\leq \frac{1}{T\gamma} \mathbb{E}[F_\delta(x_0) + h(x_0) - F_\delta(x_T) - h(x_T)] + \frac{2R}{T} \sum_{t=0}^{T-1} \mathbb{E}\big[\|\nabla F_\delta(x_t) - g_t\|\big] + \frac{2cG\sqrt{d}R^2\gamma}{\delta}
$$

$$
\overset{(a)}{\leq} \frac{1}{T\gamma} \mathbb{E}[F(x_0) + h(x_0) - F(x_T) - h(x_T) + 2\delta G] + \frac{2R}{T} \sum_{t=0}^{T-1} \sqrt{\mathbb{E}\big[\|\nabla F_\delta(x_t) - g_t\|^2\big]}
$$

$$
\quad + \frac{2cG\sqrt{d}R^2\gamma}{\delta} \tag{42}
$$

$$
\overset{(b)}{\leq} \frac{1}{T\gamma} \mathbb{E}[\phi(x_0) - \phi_{\min} + 2\delta G] + \frac{2R}{T} \sum_{t=0}^{T-1} \sqrt{\frac{16\sqrt{2\pi}dG^2}{B_t}} + \frac{cG\sqrt{d}R^2\gamma}{2\delta}
$$

$$
\overset{(c)}{\leq} R\sqrt{\frac{8cG\sqrt{d}}{T\delta} \mathbb{E}[\phi(x_0) - \phi_{\min} + 2\delta G]} + \frac{21RG\sqrt{d}}{\sqrt{B}},
$$

where (a) uses item 1 of Proposition 1, (b) uses $\phi \overset{\text{def}}{=} F + g$, $\phi_{\min} \overset{\text{def}}{=} \min_{x \in \mathbb{R}^d} \phi(x)$ and Lemma 6, (c) uses stepsize $\gamma = \frac{1}{R}\sqrt{\frac{2\delta}{TcG\sqrt{d}} \mathbb{E}[\phi(x_0) - \phi_{\min} + 2\delta G]}$ and constant batchsize $B_t \equiv B$.

Furthermore, we can select the following hyperparameters.

$$
T = \frac{32cGR^2\sqrt{d}}{\delta\epsilon^2}\big[\mathbb{E}[\phi(x_0)] - \phi_{\min} + 2\delta G\big] = \mathcal{O}(GR^2 d^{1/2}\delta^{-1}\epsilon^{-2}), \tag{43}
$$

$$
B = 1764dR^2G^2\epsilon^{-2} = \mathcal{O}(dR^2G^2\epsilon^{-2}). \tag{44}
$$

Then substituting the hyperparameters above into the convergence rate (16), we obtain the following bound, which based on item 3 of Proposition 4 implies that there exists at least one $(\delta, \epsilon)$-CGGSP in $\{x_t\}_{t=0}^{T-1}$.

$$
\min_{0 \leq t \leq T-1} \mathbb{E}[\mathcal{W}_h(x_t, \nabla F_\delta(x_t))] \leq \mathbb{E}[\mathcal{W}_h(x_{\widetilde{T}}, \nabla F_\delta(x_{\widetilde{T}}))] \leq \epsilon.
$$

## L    PROOF OF THEOREM 5

We can prove the convergence rate (17) as follows.

$$
\mathbb{E}\big[\mathcal{W}_h[x_{\widetilde{T}}, \nabla F_\delta(x_{\widetilde{T}})]\big]
$$

$$
\overset{(a)}{\leq} \frac{1}{T\gamma} \mathbb{E}[F(x_0) + h(x_0) - F(x_T) - h(x_T) + 2\delta G] + \frac{2R}{T} \sum_{t=0}^{T-1} \sqrt{\mathbb{E}\big[\|\nabla F_\delta(x_t) - g_t\|^2\big]}
$$

$$
\quad + \frac{2cG\sqrt{d}R^2\gamma}{\delta}
$$

$$
\overset{(b)}{\leq} \frac{1}{T\gamma} \mathbb{E}[\phi(x_0) - \phi_{\min} + 2\delta G] + \frac{2R}{T} \sum_{t=0}^{T-1} \sqrt{\frac{d^2G^2}{\delta^2 B_1} \sum_{j=n_t q+1}^{t} \|x_j - x_{j-1}\|^2 + \frac{16\sqrt{2\pi}dG^2}{B_0}}
$$

$$
\quad + \frac{2cG\sqrt{d}R^2\gamma}{\delta}
$$

$$
\overset{(c)}{\leq} \frac{1}{T\gamma} \mathbb{E}[\phi(x_0) - \phi_{\min} + 2\delta G] + \frac{2R}{T} \sum_{t=0}^{T-1} \sqrt{\frac{d^2G^2}{\delta^2}(2R\gamma)^2 + \frac{16\sqrt{2\pi}dG^2}{B_0}} + \frac{2cG\sqrt{d}R^2\gamma}{\delta}
$$

$$\leq \frac{1}{T\gamma} \mathbb{E}[\phi(x_0) - \phi_{\min} + 2\delta G] + \frac{GR^2\gamma}{\delta}(4d + 2c\sqrt{d}) + \frac{13RG\sqrt{d}}{\sqrt{B_0}}$$

$$\stackrel{(d)}{=} 2R\sqrt{\frac{G(4d + 2c\sqrt{d})}{T\delta} \mathbb{E}[\phi(x_0) - \phi_{\min} + 2\delta G]} + \frac{13RG\sqrt{d}}{\sqrt{B_0}},$$

where (a) uses Eq. (42) (we can see it still holds by following its proof in Appendix K), (b) uses $\phi \stackrel{\text{def}}{=} F + g$, $\phi_{\min} \stackrel{\text{def}}{=} \min_{x \in \mathbb{R}^d} \phi(x)$ and Lemma 7, (c) uses $t < (n_t + 1)q$, $B_1 = q$, $\|x_j - x_{j-1}\| = \gamma\|y_{j-1} - x_{j-1}\| \leq 2R\gamma$ (based on Eq. (15) and Lemma 8), and (d) uses stepsize $\gamma = \frac{1}{R}\sqrt{\frac{\delta \mathbb{E}[\phi(x_0) - \phi_{\min} + 2\delta G]}{TG(4d + 2c\sqrt{d})}}$.

Furthermore, we can select the following hyperparameters.

$$T = \frac{16GR^2(4d + 2c\sqrt{d})}{\delta\epsilon^2}\left[\mathbb{E}[\phi(x_0)] - \phi_{\min} + 2\delta G\right] = \mathcal{O}(GR^2 d\delta^{-1}\epsilon^{-2}), \tag{45}$$

$$B_0 = 676dR^2G^2\epsilon^{-2}, \tag{46}$$

$$B_1 = q = 26RG\epsilon^{-1}\sqrt{d}. \tag{47}$$

Then substituting the hyperparameters above into the convergence rate (16), we obtain the following bound, which based on item 3 of Proposition 4 implies that there exists at least one $(\delta, \epsilon)$-CGGSP in $\{x_t\}_{t=0}^{T-1}$.

$$\min_{0 \leq t \leq T-1} \mathbb{E}[\mathcal{W}_h(x_t, \nabla F_\delta(x_t))] \leq \mathbb{E}[\mathcal{W}_h(x_{\widetilde{T}}, \nabla F_\delta(x_{\widetilde{T}}))] \leq \epsilon.$$

# M  USE OF LARGE LANGUAGE MODELS (LLMs)

We used LLMs to generate some functions in the experimental code, and then checked and edited the code to ensure that it exactly implements the algorithms.