# OpenReview forum: "Zeroth-Order Methods for Stochastic Nonconvex Nonsmooth Composite Optimization"
_ICLR.cc/2026/Conference — Submitted to ICLR 2026_

### Official Review · Reviewer_Aq7C · 2025-10-19

**Soundness:** 3
**Presentation:** 3
**Contribution:** 2
**Rating:** 6
**Confidence:** 4

**Summary:**

This paper proposes new approximate stationary points and zeroth-order stochastic algorithms for solving the stochastic nonconvex nonsmooth composite optimization problem.

**Strengths:**

1. This paper introduces novel approximate stationary points by utilizing the Goldstein $\delta$-subdifferential for the nonsmooth stochastic composite optimization problem.

2. This paper proposes novel zeroth-order stochastic methods with an improved convergence rate than existing work [1].

**References**

[1] Liu, Z., Chen, C., Luo, L., & Low, B. K. H. (2024, July). Zeroth-order methods for constrained nonconvex nonsmooth stochastic optimization. In Forty-first International Conference on Machine Learning.

**Weaknesses:**

1. My primary concern regarding this work is that it appears to be a straightforward extension of the previous study [1], encompassing approximate stationary points and stochastic algorithms. As a result, the contribution of this work seems incremental, and its novelty is limited.

2. Since the objective function $\phi(x)$ is the sum of two nonsmooth functions, can I just apply the zeroth-order unconstrained stochastic method introduced in [2] on $\phi(x)$ to achieve a convergence rate of $\mathcal{O}(d \delta^{-1} \epsilon^{-3})$.

**References**

[1] Liu, Z., Chen, C., Luo, L., & Low, B. K. H. (2024, July). Zeroth-order methods for constrained nonconvex nonsmooth stochastic optimization. In Forty-first International Conference on Machine Learning.

[2] Kornowski, G., & Shamir, O. (2024). An algorithm with optimal dimension-dependence for zero-order nonsmooth nonconvex stochastic optimization. Journal of Machine Learning Research, 25(122), 1-14.

**Questions:**

See weakness 2.

---

> ### Author Response · Authors · 2025-11-18
> **Authors' response to Reviewer Aq7C**
>
> Q: Since the objective function $\phi(x)$ is the sum of two nonsmooth functions, can I just apply the zeroth-order unconstrained stochastic method introduced in [2] on $\phi(x)$ to achieve a convergence rate of $\mathcal{O}(d\delta^{-1}\epsilon^{-3})$?
>
> [2] Kornowski, G., \& Shamir, O. (2024). An algorithm with optimal dimension-dependence for zero-order nonsmooth nonconvex stochastic optimization. Journal of Machine Learning Research, 25(122), 1-14.
>
> A: This is an interesting observation. No, since [2] assumes Lipschitz continuous objective function but our $\phi=F+h$ may be not Lipschitz continuous with Lipschitz continuous $F$ and convex regularizer $h$.

---

> > ### Comment · Reviewer_Aq7C · 2025-11-27
> >
> > Thanks for your rebuttal! I will keep my score.

---

### Official Review · Reviewer_d6dV · 2025-10-30

**Soundness:** 4
**Presentation:** 2
**Contribution:** 2
**Rating:** 6
**Confidence:** 4

**Summary:**

This paper studies stochastic nonconvex nonsmooth composite optimization problems. The key contributions are: (i) Two new notions of approximate stationarity: ($\gamma$, $\delta$, $\epsilon$)-proximal Goldstein stationary points (PGSP) and ($\delta$, $\epsilon$)-conditional gradient Goldstein stationary points (CGGSP) which generalize Goldstein stationary points to composite objectives. (ii) Two zeroth-order algorithms: zeroth-order proximal gradient descent and zeroth-order generalized conditional gradient methods. Both methods achieve finite-time convergence guarantees to the above approximate stationary points, with and without variance reduction. (iii) Improved complexity bounds upon prior work such as Liu et al. (2024). (iv) Empirical validation on a regularized ReLU network illustrating practical convergence of both methods.

**Strengths:**

1. The definitions of PGSP and CGGSP extend Goldstein-type stationarity to nonsmooth composite settings, which had no tractable finite-time criteria before.

2. The convergence and complexity proofs are rigorous, connecting zeroth-order smoothing and nonsmooth analysis.

3. The paper clearly relates its framework to proximal methods, conditional gradient methods, and previous Goldstein-stationary notions.

4. Although small-scale, the experiments demonstrate that the methods work as claimed and variance reduction indeed accelerates convergence.

5. The exposition is organized and self-contained, with detailed assumptions, propositions, and proofs.

**Weaknesses:**

1. Only a toy ReLU network example ($d=34$) is shown. There is no comparison with baselines (e.g., stochastic subgradient, first-order PGD, or other zeroth-order methods).

2. While the theory is clean, it is not obvious how these algorithms perform in high-dimensional machine-learning applications.

3. The paper could benefit from clearer motivation and intuition before diving into technicalities.

4. The comparison to contemporary zeroth-order nonconvex optimization papers (e.g., Cutkosky 2023) could be deepened.

**Questions:**

1. Could the proposed stationarity notions be extended to settings where $h$ is nonconvex but prox-friendly?
2. How sensitive are the algorithms to the smoothing radius $\delta$ in practice?
3. Is there any connection between PGSP and the weak subgradient mappings used in Clarke’s generalized gradients?
4. Have the authors tried larger-scale tasks (e.g., CIFAR or low-rank matrix problems) to test scalability?

**Details Of Ethics Concerns:**

N/A.

---

> ### Author Response · Authors · 2025-11-18
> **Authors' response to Reviewer d6dV**
>
> Q1: Could the proposed stationarity notions be extended to settings where $h$ is nonconvex but prox-friendly?
>
> A1: For our proximal Goldstein stationary point (PGSP), yes, since it is defined on proximal mapping. For our conditional gradient Goldstein stationary point (CGGSP), no, since it requires well-defined linear minimization oracle (LMO) $\mathcal{L} _ h(g)\overset{\rm def}{=}{\arg\min} _ {y\in\mathbb{R}^d}\big[h(y)+\langle y,g\rangle\big]$, which can be guaranteed by convex $h$ but not prox-friendly $h$.
>
> Q2: How sensitive are the algorithms to the smoothing radius $\delta$ in practice?
>
> A2: When conducting experiments, we feel the algorithms are slightly sensitive to $\delta$. This can also be seen from the theory, since all the complexity results (Table 1) are proportional to $\delta^{-1}$, the stepsize $\gamma\propto\delta$ for Algorithm 1 (0-PGD), and $\gamma\propto\sqrt{\delta}$ Algorithm 2 (0-PGD).
>
> Q3: Is there any connection between PGSP and the weak subgradient mappings used in Clarke's generalized gradients?
>
> A3: Yes, our proximal Goldstein stationary point
> (PGSP) is defined using Clarke's generalized gradients. To elaborate, $x$ is a $(\gamma,\epsilon)$-PGSP if $\min _ {g\in\partial_{\delta}F(x)}\|\mathcal{G} _ {\gamma h}(x,g)\|\le\epsilon$, where $\partial _ {\delta}F(x)\overset{\rm def}{=}{\rm conv}\big[\cup _ {y\in\mathcal{B} _ d(x,\delta)}\partial F(y)\big]$ with Clark subdifferential $\partial F(y)$ (also known as Clarke's generalized gradient) defined in our Definition 1.
>
> Q4: Have the authors tried larger-scale tasks (e.g., CIFAR or low-rank matrix problems) to test scalability?
>
> A4: Experiments on Cifar-10 are shown in Appendix B.

---

> > ### Comment · Reviewer_d6dV · 2025-11-27
> > **Thank you**
> >
> > I would like to thank the authors for the response. All my questions are addressed. I will keep my score.

---

> > > ### Author Response · Authors · 2025-11-27
> > > **Thanks Reviewer d6dV**
> > >
> > > Thanks Reviewer d6dV
> > > Authors

---

### Official Review · Reviewer_1QjJ · 2025-10-31

**Soundness:** 3
**Presentation:** 3
**Contribution:** 2
**Rating:** 4
**Confidence:** 3

**Summary:**

The paper studies zeroth-order methods for stochastic nonconvex nonsmooth composite optimization, proposes PGSP and CGGSP as approximate stationary notions, and analyzes 0-PGD and 0-GCG algorithms with minibatch and variance-reduction gradient estimators, giving finite-time complexity bounds. Experiments on synthetic two-layer ReLU and ResNet-20 validate algorithmic behavior.

**Strengths:**

- The authors extend the notion of constrained stationarity to general nonsmooth composite settings by replacing the projection operator with a proximal mapping. This yields a unified definition applicable to many practical problems. Two zeroth-order algorithms are proposed to handle different oracle settings (proximal vs LMO). The convergence analyses and intractability results are presented clearly.
- The paper is generally well-written, logically organized, and easy to follow. Definitions and assumptions are stated clearly, and the appendix provides detailed proofs.

**Weaknesses:**

-  While the definition of PGSP extends previous constrained stationary notions to the nonsmooth setting, this extension is **rather straightforward** — essentially replacing the projection operator in the constrained case by a proximal operator. Similarly, the algorithmic framework closely follows that of **Liu et al. (2024)** for the constrained Lipschitz case, with minor modifications.
- Although the paper claims an improvement in complexity bounds, the improvement is only in the **parameter dependence** , which is mainly due to the tight bound for $F_\delta(x_0) - F_\delta(x_T)$ (as mentioned in **Comparison with Constrained Optimization**), while the overall order of complexity remains identical.  For the tight bound, I think the new term $\psi - \psi_*$ introduced in the analysis may weaken the claimed improvement, and it is unclear whether this scaling is indeed tight or essential. The resulting theory, though consistent, does not introduce fundamentally new mathematical tools or algorithmic ideas. A detailed side-by-side comparison of assumptions, complexities, and definitions would strengthen the contribution.
-  The complexity bounds in Table 1 do not explicitly include $\gamma$. If the complexity is indeed independent of $\gamma$, does that imply the same rate holds for any $\gamma$? This point needs further explanation, as $\gamma$ appears both in the proximal mapping and smoothing radius, and typically affects the variance–bias trade-off.

**Questions:**

Please clarify whether the derived iteration and query complexities depend on $\gamma$.
 If not, why does $\gamma$ appear in the proximal update?
 Intuitively, the step size affects both convergence and stationarity precision — this should be explicitly reflected in the bounds.

$\delta$ appears both in the smoothing process and the stationarity definition. How should δ be chosen in practice? What happens when δ is too small — does the variance term blow up?

---

> ### Author Response · Authors · 2025-11-18
> **Authors' response to Reviewer 1QjJ**
>
> Weakness 2: Although the paper claims an improvement in complexity bounds, the improvement is only in the **parameter dependence**, which is mainly due to the tight bound for $F _ {\delta}(x_0)-F _ {\delta}(x_T)$ (as mentioned in **Comparison with Constrained Optimization**), while the overall order of complexity remains identical. For the tight bound, I think the new term introduced in the analysis may weaken the claimed improvement, and it is unclear whether this scaling is indeed tight or essential. The resulting theory, though consistent, does not introduce fundamentally new mathematical tools or algorithmic ideas. A detailed side-by-side comparison of assumptions, complexities, and definitions would strengthen the contribution.
>
> A2: We prove that our bound is tighter as follows. (Liu et al., 2024) uses the bound $F_{\delta}(x_0)-F_{\delta}(x_T)\le 2GR$ on the constrained optimization problem where the constraint set $\Omega$ has radius $R$. In contrast, in the special case of constrained optimization, our bound $\mathbb{E}[\phi(x_0)]-\phi_{\min}+2\delta G$ can be further upper bounded by $\mathbb{E}[\phi(x_0)]-\phi_{\min}+2\delta G\le(2R+2\delta)G\le 4RG$ since $\phi(x)=F(x)$ is $G$-Lipschitz continuous in $\Omega$ and $\delta\le R$. Therefore, our bound in the worst case has the same order as their bound $2GR$, but may also have lower order (i.e. $\phi(x_0)-\phi^*\ll 2GR$) for some good initializations $x_0$. Moreover, on unbounded set without such radius $R$, their bounds do not hold but our bound for our Algorithm 1 (with proximal mapping) still holds.
>
> Comparison on assumptions: Our Algorithm 1 on Lipschitz continuity is the same as Assumption 3.2 of (Liu et al., 2024), while our Assumptions 2 and 3 on the convex regularizer $h$ generalize Assumption 3.1 of (Liu et al., 2024) that $h(x)=0$ in a convex and compact set $\Omega$ but $+\infty$ outside $\Omega$. This has been illustrated after our Assumption 3.
>
> Comparison on complexities: As mentioned above, all our complexities become tighter than the corresponding ones in (Liu et al., 2024), by replacing $2RG$ with $\mathbb{E}[\phi(x_0)]-\phi_{\min}+2\delta G$. This has been illustrated after our Theorems 2, 3 and 5.
>
> Comparison on definitions: When restricted to constrained optimization problem, our definitions of PGSP and CGGSP respectively reduce to those of the generalized Goldstein stationary point and the Clarke Frank-Wolfe stationary point in (Liu et al., 2024), as illustrated right after Definitions 5 and 7 respectively.
>
> Weakness 3: The complexity bounds in Table 1 do not explicitly include
> $\gamma$. If the complexity is indeed independent of $\gamma$, does that imply the same rate holds for any $\gamma$? This point needs further explanation, as $\gamma$ appears both in the proximal mapping and smoothing radius, and typically affects the variance–bias trade-off.
>
> A3: See our answer to Q1 below.
>
> Q1: Please clarify whether the derived iteration and query complexities depend on $\gamma$. If not, why does $\gamma$ appear in the proximal update? Intuitively, the step size affects both convergence and stationarity precision — this should be explicitly reflected in the bounds.
>
> A1: In all our convergence theorems (Theorems 2-5), we select the stepsize $\gamma$ which depends on $\delta,d$, so the dependency of the complexities on $\gamma$ becomes dependency on $\delta,d$.
>
> Q2: $\delta$ appears both in the smoothing process and the stationarity definition. How should $\delta$ be chosen in practice? What happens when $\delta$ is too small — does the variance term blow up?
>
> A2: Good question. $\delta$ can be fine-tuned in practice. Note that the zero-th order stochastic gradient approximates $\nabla F_{\delta}(x)$ which is $\frac{cG\sqrt{d}}{\delta}$ Lipschitz continuous, so too small $\delta$ can make $\nabla F_{\delta}(x)$ changing faster with $x$, so we need to tune down the stepsize $\gamma$ which slows down convergence. This is reflected in Table 1 where the complexities for all the algorithms are proportional to $\delta^{-1}$.

---

### Meta-Review · Area_Chair_SEfy · 2025-12-25

**Summary:**

This paper studies zeroth-order methods for problems whose objective is a sum of two nonconvex and nonsmooth functions. New notions of approximate stationary point are defined, and iteration complexity for obtaining such approximate solution is analyzed. The paper has the following issues as pointed out by the reviewers: (i) the algorithm and contribution are incremental compared to the existing work (Liu et al. 2024); (ii) the applicability is limited, and in the numerical experiments, only one example is tested. The author’s rebuttal doesn’t address these issues satisfactorily.

**Reviewer Concerns:**

The reviewers raised the following concerns: (i) the contribution is incremental comparing with existing work; (ii) the applicability of the proposed algorithm is limited. These issues are still outstanding.

**Reviewer Scores:**

see above. It is not likely the reviewers will increase their scores due to the issues that are still outstanding.

---

### Decision · Program_Chairs · 2026-01-26

Reject